# Werner syndrome exonuclease promotes gut regeneration and causes age-associated gut hyperplasia in *Drosophila*

Kun Wu[1☉], Juanyu Zhou[1☉], Yiming Tang[1☉], Qiaoqiao Zhang[1☉], Lishou Xiong[2☉], Xiaorong Li[3], Zhangpeng Zhuo[4], Mei Luo[2], Yu Yuan[1], Xingzhu Liu[1], Zhendong Zhong[4], XiaoXin Guo[1], Zihua Yu[1], Xiao Sheng[1], Guanzheng Luo[4], Haiyang Chen [1]*

1 West China Centre of Excellence for Pancreatitis and Laboratory of Metabolism and Aging, Frontiers Science Center for Disease-related Molecular Network, State Key Laboratory of Respiratory Health and Multimorbidity and National Clinical Research Center for Geriatrics, West China Hospital, Sichuan University, Chengdu, China, 2 Department of Gastroenterology and Hepatology, The First Affiliated Hospital of Sun Yat-sen University, Guangzhou, China, 3 School of Public Health (Shenzhen), Shenzhen Campus of Sun Yat-sen University, Shenzhen, Guangdong, China, 4 Key Laboratory of Gene Engineering of the Ministry of Education, State Key Laboratory of Biocontrol, School of Life Sciences, Sun Yat-sen University, Guangzhou, Guangdong, China

☉ These authors contributed equally to this work.
* chenhy82@scu.edu.cn

## Abstract

Human Werner syndrome (adult progeria, a well-established model of human aging) is caused by mutations in the Werner syndrome (WRN) gene. However, the expression patterns and functions of WRN in natural aging remain poorly understood. Despite the link between WRN deficiencies and progeria, our analyses of human colon tissues, mouse crypts, and *Drosophila* midguts revealed that WRN expression does not decrease but rather increases in intestinal stem cells (ISCs) with aging. Mechanistically, we found that the *Drosophila* WRN homologue (WRNexo) binds to Heat shock 70-kDa protein cognate 3 (Hsc70-3/Bip) to regulate the unfolded protein response of the endoplasmic reticulum ($UPR^{ER}$). Activation of the WRNexo-mediated $UPR^{ER}$ in ISCs is required for ISC proliferation during injury repair. However, persistent DNA damage during aging leads to chronic upregulation of WRNexo in ISCs, where excessive WRNexo-induced ER stress drives age-associated gut hyperplasia in *Drosophila*. This study reveals how elevated WRNexo contributes to stem cell aging, providing new insights into organ aging and the pathogenesis of age-related diseases, such as colon cancer.

## Introduction

Human progeria syndromes are progressive disorders that lead to the appearance of unusually accelerated aging and a shortened life span. Werner syndrome, characterized by accelerated aging after puberty, is a representative type of progeria. This

**Data availability statement:** All data presented are available in the main text or the supplementary materials. The RNA-seq data supporting this study have been deposited in PRJNA944585 (Fig 4A, 4B), with source data provided in S1 and S2 Tables.

**Funding:** This work was supported by the National Key R&D Program of China (2020YFA0803602), the National Natural Science Foundation of China (32470879 and 92157109) (H.C.), Sichuan Science and Technology Program, Sichuan Provincial Natural Science Foundation (Grant NO. 2025ZNSFSC0720 (H.C.)), and the 1·3·5 project for disciplines of excellence, West China Hospital, Sichuan University (ZYYC20024) (HC) and 1·3·5 projects for Artificial Intelligence (ZYAI24024), West China Hospital, Sichuan University. The funders had no role in study design, data collection and analysis, decision to publish, or preparation of the manuscript.

**Competing interests:** The authors have declared that no competing interests exist.

**Abbreviations :** ACI, after clone induction; BDSC, Bloomington *Drosophila* Stock Center; BLM, bleomycin; *Bsk*, *Basket*; CA, constitutively active; *Cat*, *Catalase*; DDR, DNA damage response; DEPC, diethylpyrocarbonate; DHE, dihydroethidium; Dl⁺, Delta⁺; DN, dominant-negative; DR, dietary restriction; DSBs, DNA double-strand breaks; EBs, enteroblasts; ECs, enterocytes; EEs, enteroendocrine cells; ER, endoplasmic reticulum; *Hep*, *Hemipterous*; Hsc70-3/ Bip, Heat shock 70-kDa protein cognate 3; IP, immunoprecipitation; ISCs, intestinal stem cells; JNK, c-Jun N-terminal kinase; MARCM, mosaic analysis with a repressible cell marker; MS, mass spectrometry; MSI, microsatellite instability; PBS, phosphate-buffered saline; PQ, paraquat; Pros⁺, Prospero⁺; PVDF, polyvinylidene difluoride; RNA-seq, RNA-sequencing; ROI, region of interest; ROS, reactive oxygen species; RT-qPCR, real-time quantitative PCR; THFC, TsingHua Fly Center; TM, tunicamycin; UPRᴱᴿ, unfolded protein response of the endoplasmic reticulum; VDRC, Vienna Drosophila Resource Center; WRN, Werner syndrome; WRNexo, Werner syndrome exonuclease; Xbp1ˢ, spliced X-box binding protein 1.

disease is caused by mutations in the *WRN* gene, which encodes the WRN protein with both exonuclease and RecQ helicase activities [1]. In patients with Werner syndrome, mutations have been identified in both the exonuclease and helicase domains [1–3]. Individuals with Werner syndrome die prematurely and exhibit age-related degenerative disorders, including skin ulcers, cataracts, atherosclerosis, diabetes, osteoporosis, subcutaneous fat loss, and cancers, early in life [4]. Studies of cultured cells from patients with Werner syndrome and in animal models harboring *WRN* mutations have shown that WRN plays roles in DNA replication, DNA damage repair, transcription, and recombination as well as telomere maintenance [1,5,6]. Despite extensive studies of the link between WRN and progeria phenotypes, its physiological function in natural aging remains largely unexplored.

By studying colonic and gastric cancer cells, two recent studies have shown that WRN is upregulated in cancers with microsatellite instability (MSI) and is essential for survival in these cancers [7,8]. However, the physiological function of WRN in normal gut homeostasis and age-associated gastrointestinal tumorigenesis has yet to be explored. The progressive loss of proliferative balance is a widely known hallmark of intestinal epithelium aging, leading to age-associated diseases, such as colorectal cancers [9,10]. The long-term homeostasis of the human intestinal epithelium relies on intestinal stem cells (ISCs), which are long-lived and possess unlimited potential to replicate and replenish old or damaged epithelial cells. Thus, a better understanding of WRN function in ISCs could provide insight into the physiology of gut homeostasis and aging.

The *Drosophila* adult midgut is a well-established system to study stem cell biology in tissue homeostasis, regeneration, and aging. Similar to the human intestine, the *Drosophila* midgut epithelium is sustained by resident ISCs that are long-lived due to their capacity for self-renewal [11–14]. *Drosophila* ISCs (specifically expressing the Notch ligand Delta) are scattered along the basement membrane of the midgut epithelium, where they divide to self-renew and produce non-dividing enteroblasts (EBs, which further differentiate into absorptive enterocytes (ECs)) or enteroendocrine mother cells (which further differentiate into secretory enteroendocrine cells (EEs)) [11,12,15–18] (S1A Fig). Under normal homeostatic conditions, most ISCs are in a quiescent state and divide slowly. In response to gut epithelium damage, these ISCs are rapidly activated with transiently increased proliferation rate to initiate EC and EE production. This regeneration process is regulated by multiple stress and mitogenic signaling pathways, such as epidermal growth factor receptor [19,20], c-Jun N-terminal kinase (JNK) [21], Janus kinase-signal transducer and activators of transcription [22,23], insulin [24–26], Hippo [27], Wingless [28,29], and bone morphogenetic protein signaling pathways [30].

During aging, *Drosophila* develops epithelial hyperplasia in midguts that results from uncontrolled ISC over-proliferation and deficient EB differentiation [21,31]. This age-associated hyperplasia is associated with the chronic activation of diverse stress signaling mechanisms, including inflammatory conditions, metabolic changes, oxidative stress, and endoplasmic reticulum (ER) stress [32–35]. The unfolded protein response of the endoplasmic reticulum (UPRᴱᴿ) which adjusts the protein folding

capacity in response to stress, has emerged as a central regulator of *Drosophila* midgut epithelial homeostasis during regeneration and aging [36,37]. And excessive ER stress, in coordination with oxidative stress and JNK signaling, drives ISC hyper-proliferation in aging flies [36]. Although it has been linked to age-associated gut hyperplasia in old *Drosophila*, the underlying causes of the chronic UPR$^{ER}$ activation during aging remain poorly understood.

Unlike human WRN, which has both exonuclease and helicase activities, *Drosophila* WRN possesses only exonuclease activity and is therefore named Werner syndrome exonuclease (WRNexo). WRNexo shares 35% identity and 59% similarity with the exonuclease domain of human WRN [38]. Similar to WRN-deficient human cells, *Drosophila* *WRNexo*-null mutants exhibit heightened DNA damage and increased sensitivity to DNA replication stress [38–42].

Using the *Drosophila* midgut as a model system, we discovered that age-associated genomic instability induces WRNexo upregulation in ISCs. Elevated WRNexo exacerbates ER stress in ISCs by physically interacting with a UPR$^{ER}$ sensor, Heat shock 70-kDa protein cognate 3 (Hsc70-3/Bip) [43,44], and suppressing its activation. Increased ER stress drives ISC hyperplasia through reactive oxygen species (ROS) and JNK signaling. Furthermore, our study demonstrates that the age-associated upregulation of WRNexo is directly triggered by elevated genomic instability in aged *Drosophila* ISCs. Our findings underscore the critical role of WRN in maintaining gut epithelial homeostasis, regulating natural aging, and involving age-associated hyperplasia in mammals.

## Results

### WRN is highly expressed in intestinal stem cells and upregulated upon natural aging

To investigate the functions of WRN in the human gut during natural aging, we analyzed normal colon tissues from both young and old individuals. Given that WRN mutations cause Werner syndrome—a condition characterized by premature aging—we anticipated that WRN expression would decrease with age. However, immunofluorescence microscopy revealed that, contrary to our prediction, WRN expression actually increased dramatically in the colon crypts of elderly individuals (Fig 1A–1C). Western blotting and real-time quantitative PCR (RT-qPCR) analyses confirmed that WRN protein and mRNA expression levels were higher in the colon tissue from elderly individuals compared to young individuals (Fig 1D–1F). Recent studies have shown that WRN expression is upregulated in cultured colonic and gastric cancer cell lines, promoting survival in colorectal cancers [7,8]. Since colon carcinoma is an age-associated disease, we further analyzed WRN expression in human colon carcinoma. Immunofluorescence microscopy revealed significantly higher WRN expression in the crypts of colorectal cancer regions compared to adjacent normal tissues (Fig 1G–1I). RT-qPCR analyses confirmed that *WRN* expression was elevated in colon carcinoma tissues (Fig 1J). Additionally, we analyzed WRN expression in aged mice. Both western blotting and RT-qPCR analyses demonstrated elevated WRN expression in the crypts of aged mice (Fig 1K–1M). These findings suggest that the increased WRN expression observed in elderly colon tissues and cancerous tissues reflects distinct functional roles of WRN in natural tissue aging and in premature aging associated with Werner syndrome, which is characterized by a WRN deficiency.

Given the elevated expression of WRN in intestinal crypts of elderly individuals and mice, we further investigated whether WRN plays a role in gut homeostasis and natural aging using the *Drosophila* midgut system [11,12,45]. *Drosophila* WRNexo (CG7670) has been identified as an ortholog of human WRN exonuclease for modeling Werner syndrome [38,46]. To trace endogenous WRNexo expression during gut homeostasis and aging in *Drosophila*, we generated the *WRNexo-mCherry* reporter line using the CRISPR/Cas9 knock-in system (S1B Fig). Although expressed throughout the midgut, WRNexo was more highly expressed in the posterior region (R4–R5) compared to the anterior region (R1–R3) (S1C and S1D Fig). Consistent with observations in human intestines, WRNexo expression levels in the midguts of aged *Drosophila* were significantly higher than those in young flies (Figs 1N–1P and S1C and S1D). RT-qPCR analyses revealed that the elevation of *WRNexo* expression in the midguts of aged flies might be explained by an increase in transcription (Fig 1Q). Analysis of previously published RNA-sequencing (RNA-seq) data from whole guts from young and old flies [47], yielded similar results, showing elevated WRNexo expression in aged guts (S1E and S1F Fig).

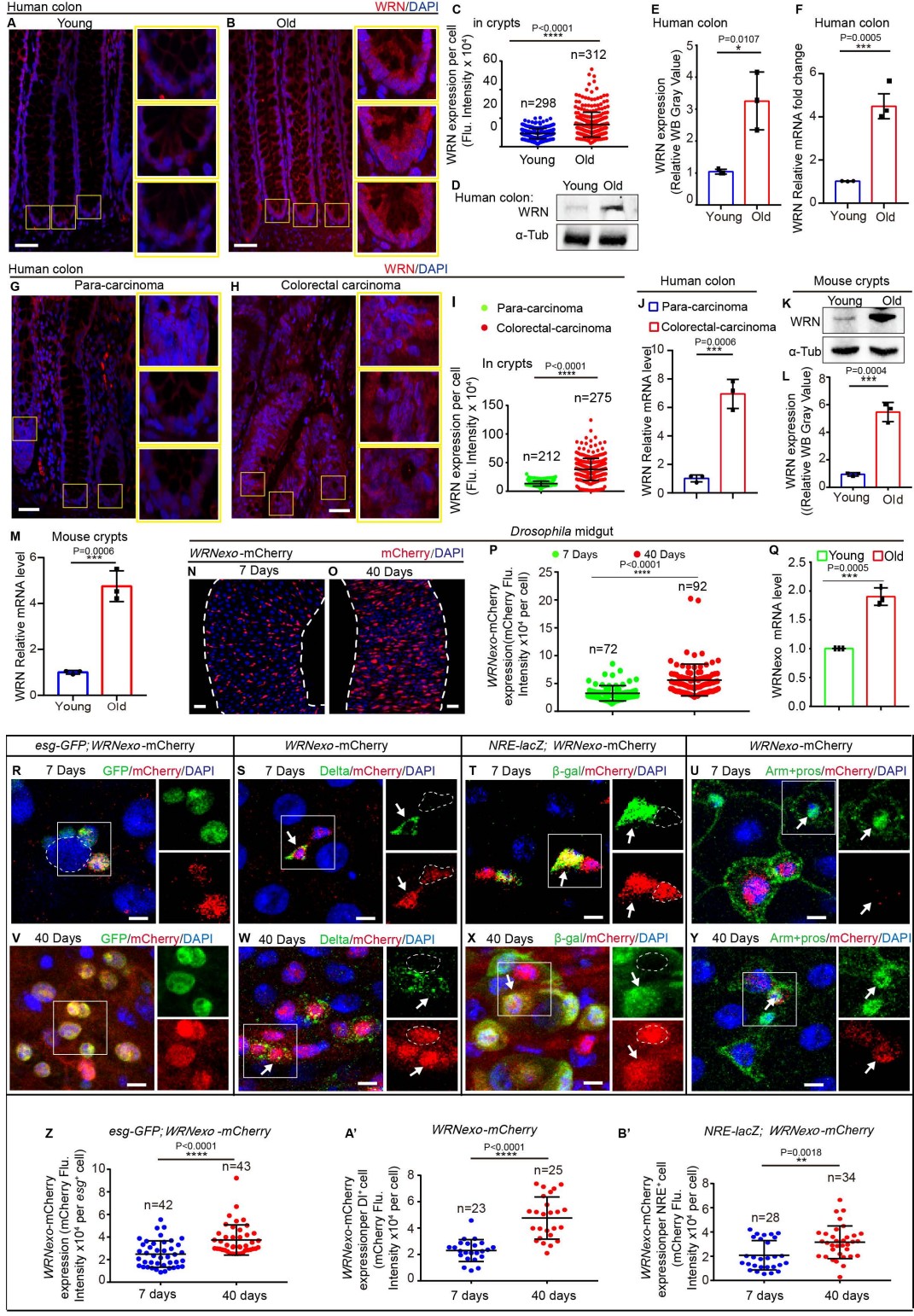

**Fig 1. WRN is highly expressed in intestinal stem cells and upregulated upon natural aging. (A, B)** Immunofluorescence staining of WRN in human para-carcinoma colon tissues from young (the mean age was 29 years, $n$ = 4) (A) and old individuals (the mean age was 67 years, $n$ = 7) **(B)**. WRN was detected using a specific antibody (red). Boxed areas in (A, B) are enlarged to the right. **(C)** Quantification of WRN fluorescence intensity

in the boxed regions of crypts from panels (A, B). Colon samples were obtained from four young patients and seven old patients. Each dot represents one cell. **(D)** Western blotting results indicate an increase in WRN expression in colons with aging. Loading controls, α-Tubulin. Colon samples were from four young patients and seven old patients. **(E)** Quantification of WRN band intensity from experiments shown in panel (D). **(F)** Relative mRNA fold changes of *WRN* in normal young and old para-carcinoma colons. Data are normalized to young colons (control, set to 1). Error bars indicate the standard deviation (SD) from three independent experiments. **(G, H)** Representative images of WRN staining in human colons. Normal para-carcinoma colons (as control, G) and carcinoma colons from patients with colon cancer (H). Cells were stained by WRN (red). The boxed areas in (G, H) are enlarged to the right. **(I)** Quantification of WRN fluorescence intensity in experiments (G, H). Each dot represents one cell in boxed regions of crypts. **(J)** Relative mRNA fold changes of *WRN* in para-carcinoma colons and carcinoma colons. Data are normalized to para-carcinoma colons (control, set to 1). Error bars indicate the standard deviation (SD) from three independent experiments. **(K)** Western blotting showing WRN expression levels in crypts from young and old mouse colons. Loading controls, α-Tubulin. **(L)** Quantification of WRN band intensity from experiments shown in panel (K). **(M)** Relative mRNA fold changes of *WRN* in young and old mouse crypts. Data are normalized to young crypts (control, set to 1). Error bars indicate the standard deviation (SD) from three independent experiments. **(N, O)** Representative images of *WRNexo*-mCherry expression in young (N) and old (O) *Drosophila* midguts. **(P)** Quantification of mCherry fluorescence intensity per cell from experiments (N, O); each dot represents one *WRNexo*-mCherry$^+$ cell. **(Q)** Relative mRNA fold changes of *WRNexo* in young and old *Drosophila* midguts. Data are normalized to young midguts (control, set to 1). Error bars indicate the standard deviation (SD) from three independent experiments. **(R–U)** Immunofluorescence images of *WRNexo*-mCherry expression in young (7-day-old) *Drosophila* posterior midguts. WRNexo-mCherry was detected in *esg*-GFP$^+$ cells (R), Dl$^+$ ISCs (white arrow) (S), and NRE$^+$ EBs (white arrow) (T) but not in Pros$^+$ EEs (white arrow) (U). Dashed lines indicate only *WRNexo*-mCherry-positive cells (S and T). Dashed lines indicate ECs (R). **(V–Y)** Immunofluorescence images of WRNexo-mCherry in posterior midguts of aged (40-day-old) *Drosophila*. WRNexo-mCherry expression increased in *esg*-GFP$^+$ cells (V), Dl$^+$ ISCs (white arrow) (W), NRE$^+$ EBs (white arrow) (X), and some Pros$^+$ EEs (white arrow) (Y). Dashed lines indicate only WRNexo-mCherry-positive cells (W and X). **(Z, A′, B′)** Quantification of mCherry fluorescence intensity in *esg*-GFP$^+$ (Z), Dl$^+$ (A′), and NRE$^+$ (B′) cells from young and aged posterior midguts. Each dot represents one cell. DAPI-stained nuclei (blue). Scale bars represent 50 μm in A, B, and G, H, 25 μm in N and O, 5 μm in R to U and V to Y. Error bars represent SD. Student *t* tests, *$p < 0.05$, **$p < 0.01$, ***$p < 0.001$, ****$p < 0.0001$, and NS (non-significant) represents $p > 0.05$. See also S1 Fig and S3 Table. Underlying data and statistical analysis in S1 Data.

Importantly, we found that WRNexo was not ubiquitously expressed in the *Drosophila* intestinal epithelium but was specifically expressed in cells with a relatively small nucleus. By co-staining with cell type-specific markers, in the posterior midguts, we found that WRNexo expression in the intestinal epithelium of young *Drosophila* was restricted to Delta$^+$ (Dl$^+$) ISCs and NRE$^+$ EBs (both ISCs and EBs are *esg*-GFP$^+$, Fig 1R–1T). WRNexo expression was almost undetectable in terminally differentiated ECs (*esg*-negative cells with a polyploid nucleus; Fig 1R) and Prospero$^+$ (Pros$^+$) EEs (Fig 1U). Furthermore, WRNexo expression was significantly higher in the ISCs and EBs of aged flies (Fig 1V–1X and 1Z–1B′) compared to young flies (Fig 1R–1T and 1Z–1B′). In the midguts of aged flies, WRNexo was expressed in a subset of Pros$^+$ EEs (Fig 1Y), which may result from the accumulation of *esg*$^+$ premature EEs in aged midguts [16,17,21]. Additionally, in the anterior midguts, the expression pattern of WRNexo was similar to that observed in the posterior midguts (S1G–S1J Fig). These data suggest that WRNexo may function in intestinal stem and progenitor cells in *Drosophila*. Beyond the intestines, WRNexo was found to be highly expressed in female and male germline stem and progenitor cells in *Drosophila* (S1K and S1L Fig).

## WRNexo is essential for the maintenance of gut homeostasis and the activation of ISC proliferation after injury

To study the physiological function of WRNexo in tissue homeostasis and natural aging, we generated *WRNexo* mutant *Drosophila* lines using the CRISPR/Cas9 knock-out system (S2A Fig). We obtained three alleles (*WRNexo$^{10}$*, *WRNexo$^{12}$*, and *WRNexo$^{15}$*) carrying deletions in *WRNexo* exon 1 or exon 2 (S2A Fig). Since the coding region of *WRNexo* is almost entirely disrupted in these alleles, they can be regarded as null alleles. *WRNexo-null* (*WRNexo$^{10/12}$* and *WRNexo$^{10/15}$* transheterozygotes) mutant flies were viable. The overall morphology of midguts (Figs 2A, and S2B–S2E) and the proliferation rate of ISCs, as indicated by immunostaining for pH3 (a marker of ISC proliferation) and *esg*-GFP (a marker indicating the total number of ISCs and EBs), were largely similar in young (7 days old) *WRNexo-null* mutant flies and wild-type flies (S2F–S2J Fig). However, in the midguts of 40-day-old *WRNexo-null* flies (Fig 2C), epithelial cells were more sparsely distributed compared to those in wild-type controls (Fig 2B and 2C). The midgut length was significantly shorter in aged *WRNexo-null* flies (40 days old) compared to wild-type flies of the same age (Fig 2A, 2D, and 2E). Additionally, while the numbers of pH3$^+$ cells and *esg*-GFP$^+$ cells increased in the midguts of old wild-type flies due to the accumulation of ISCs

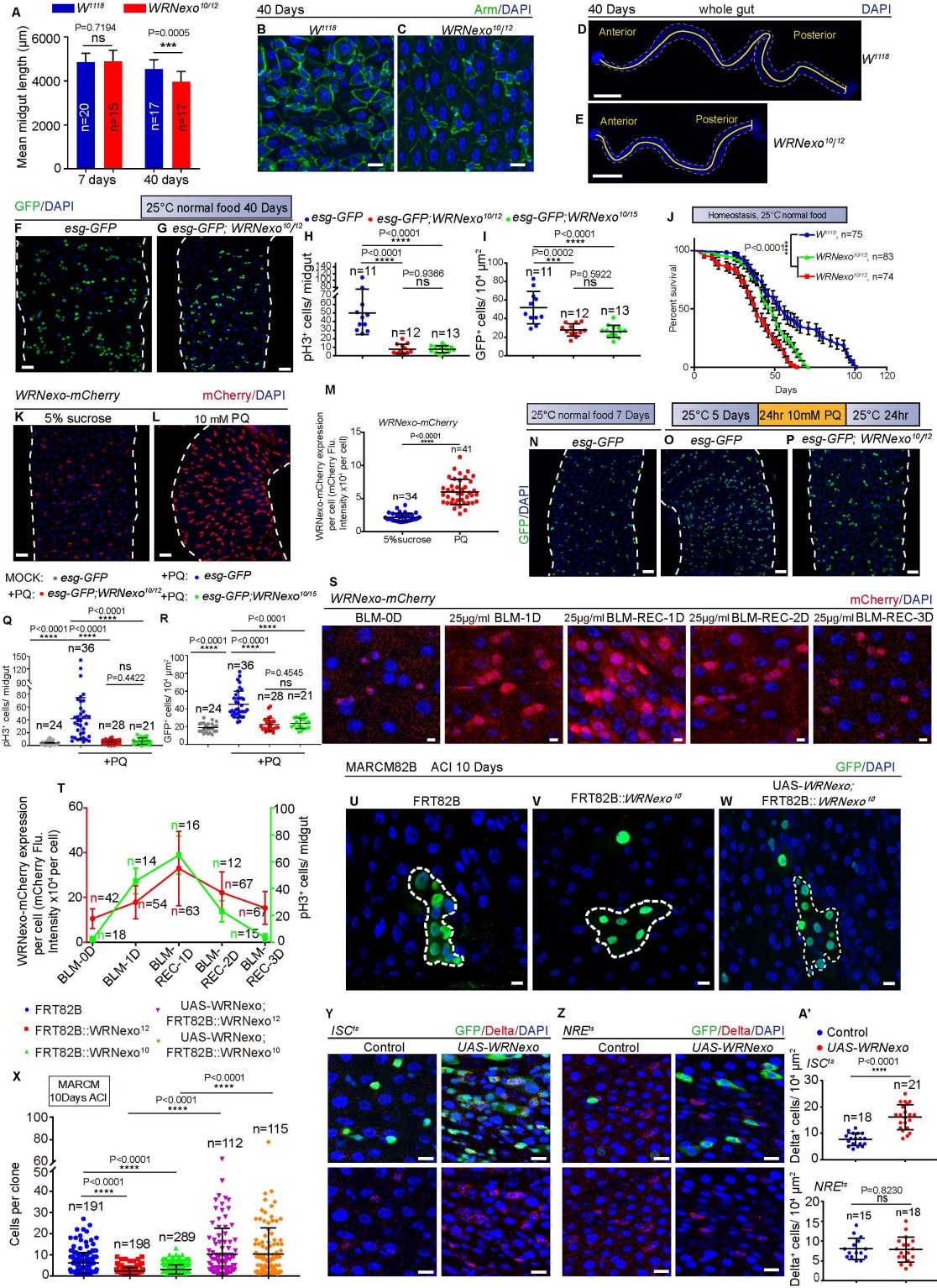

**Fig 2. WRNexo is essential for maintaining gut homeostasis and for activating ISC proliferation after injury. (A)** Mean length of young and old midguts from flies with indicated genotypes. Lengths were measured from images of DAPI-stained guts, spanning from the midline of the proventriculus to the midgut–hindgut boundary. *n* is as indicated. **(B, C)** Representative images of DAPI and Armadillo staining in midguts from 40-day-old flies.

Epithelial cells in the *WRNexo-null* midguts **(C)** were sparsely distributed compared to those in wild-type flies (B). **(D, E)** DAPI-stained whole midgut images. *WRNexo-null* midguts (E) were significantly shorter than wild-type midguts (D) in 40-day-old flies. **(F, G)** Immunofluorescence images of esg-GFP midguts in aged (40 days) control (*esg-GFP/+*, F) and *WRNexo-null* flies (*esg-GFP/+; WRNexo¹⁰/WRNexo¹²*, G). **(H)** Quantification of the pH3⁺ cells in 40-day-old midguts from control flies (*esg-GFP/+*) and *WRNexo-null* flies (*esg-GFP/+; WRNexo¹⁰/WRNexo¹²* and *esg-GFP/+; WRNexo¹⁰/WRNexo¹⁵*). Each dot represents one midgut, *n* is as indicated. **(I)** Quantification of *esg*-GFP⁺ cells in a 10,000 µm² area of midguts with indicated genotypes from experiments (F, G). Each dot represents a region of interest (ROI) (region of interest). ROI size was 10,000 µm², *n* is as indicated. **(J)** Survival rate of *WRNexo-null* flies raised on normal food and temperature conditions. WT (*W¹¹¹⁸*) flies served as controls. **(K, L)** Immunofluorescence images of *WRNexo*-mCherry reporter midguts treated with 5% sucrose (K) or paraquat (PQ) for 1 day, followed by a 1-day recovery on normal food (hereafter referred as PQ-REC-1D) (L). **(M)** Quantification of mCherry fluorescence intensity in midguts from flies treated with 5% sucrose (K) or PQ-REC-1D (L). **(N–P)** Immunofluorescence images of *esg*-GFP midguts from control flies under normal conditions (N), PQ-REC-1D-treated control (O), and *WRNexo-null* (P) flies. **(Q)** Quantification of pH3⁺ cells in whole midguts from the experiment (N–P). Each dot represents one midgut, *n* is as indicated. **(R)** Quantification of *esg*-GFP⁺ cell numbers in a 10,000 µm² area of midguts with indicated genotypes from experiments (N–P). Each dot represents a ROI. ROI size was 10,000 µm², *n* is as indicated. **(S)** *WRNexo*-mCherry expression during bleomycin (BLM)-induced regeneration. Flies were cultured under normal conditions (BLM-0D), exposed to BLM for one day (BLM-1D), or recovered on regular food for 1–3 days (BLM-REC-1D to BLM-REC-3D). **(T)** Line graphs depicting trends in mCherry fluorescence intensity (red line) and pH3⁺ cell counts (green line) during BLM-induced regeneration. *n* is as indicated. **(U–W)** Immunofluorescence images of FRT82B control (U), *WRNexo-null* (V), and *WRNexo-null* with *WRNexo* overexpression (W) in MARCM clones 10 days after clone induction (ACI). **(X)** Quantification of MARCM clone size in FRT82B control, *WRNexo-null*, and *WRNexo-null* with *WRNexo* overexpressing MARCM clones 10 days ACI. Each dot represents a clone, *n* is as indicated. **(Y)** Immunofluorescence images of midguts carrying *ISCᵗˢ* (*esg-GAL4, UAS-GFP; tub-Gal80ᵗˢ, NRE-Gal80*) (Control) and *WRNexo* cDNA overexpression vectors. The midguts were stained with GFP and Delta. Delta and DAPI channel images are shown in the lower panel. **(Z)** Immunofluorescence images of midguts carrying *NREᵗˢ* (*NRE-GAL4, UAS-GFP, tub-Gal80ᵗˢ*) (Control) and *WRNexo* cDNA overexpression vectors. The midguts were stained with GFP and Delta. The Delta and DAPI channel images are shown in the lower panel. **(A′)** Quantification of Delta⁺ cells per ROI (10,000 µm²) in midguts from experiments in (Y, Z). Each dot represents a ROI. The ROI size was 10,000 µm², *n* is as indicated. DAPI-stained nuclei (blue). Scale bars represent 5 µm in S, 10 µm in B, C, U–W, Y, and Z, 25 µm in F, G, K, L, and N–P, 200 µm in D and E. Error bars represent SD. Student *t* tests, *$p < 0.05$, **$p < 0.01$, ***$p < 0.001$, ****$p < 0.0001$, and NS (non-significant) represents $p > 0.05$. See also S2 Fig and S3 Table. Underlying data and statistical analysis in S2 Data.

and EBs [21,35,48], these counts remained low in *WRNexo-null* flies (Fig 2F–2I). In addition, we found that *WRNexo-null* flies showed a significantly shorter life span than that of wild-type flies (Fig 2J). These data indicate that the *WRNexo* depletion disrupts gut homeostasis during aging in *Drosophila*, as evidenced by shortened gut length, impaired ISC proliferation, compromised epithelial integrity, and reduced life span.

The accumulation of damage in response to diverse environmental stimuli is thought to contribute to the aging process in animals [35]. In adult *Drosophila*, resident ISCs in the midgut play a crucial role in maintaining gut homeostasis by replenishing damaged epithelial cells and promoting tissue repair after injury [49]. This regenerative response is essential for sustaining intestinal homeostasis. To study ISC behavior under stress, researchers commonly employ various stressors, such as paraquat (PQ) or bleomycin (BLM), in the *Drosophila* midgut [49,50]. Interestingly, we found that WRNexo protein expression increased significantly in the midguts of *Drosophila* after PQ-induced injury (Fig 2K–2M), suggesting that WRNexo may play a role in the cellular response to injury. Consistent with our hypothesis, we observed that the pH3⁺ and *esg*-GFP⁺ cell counts in wild-type midguts increased rapidly along the epithelium to repair PQ-induced intestinal damage [21,34] (Fig 2N, 2O, 2Q, and 2R). However, in the *WRNexo-null* midguts, the numbers of pH3⁺ cells and *esg*-GFP⁺ cells does not increase after injury (Fig 2P–2R), indicating impaired ISC proliferation in *WRNexo*-depleted *Drosophila*. Furthermore, both WRNexo expression and ISC proliferation (as indicated by the presence of pH3⁺ cells) followed a similar pattern during the midgut regeneration induced by BLM. Initially, WRNexo expression and ISC proliferation increased, followed by a gradual decrease (Fig 2S and 2T). These findings suggest that the upregulation of WRNexo during midgut regeneration is crucial for the activation of ISC proliferation.

To further demonstrate that WRNexo is essential for the activation of ISC proliferation, we analyzed *WRNexo-null* clones using a mosaic analysis with a repressible cell marker (MARCM), which labels all progeny of a single active ISC with a visible GFP marker. Consistently, *WRNexo-null* clones were much smaller than wild-type clones (Fig 2U, 2V, and 2X). Importantly, the ISC proliferation defect observed in the *WRNexo-null* clones could be fully reversed by expressing *WRNexo* cDNA (Fig 2W and 2X). In addition, *esg-Gal4*-driven Flip-out (F/O) lineage tracing combined with RNAi showed that the depletion of *WRNexo* led to the ISC proliferation defect (S2K–S2N Fig). Two *WRNexo*

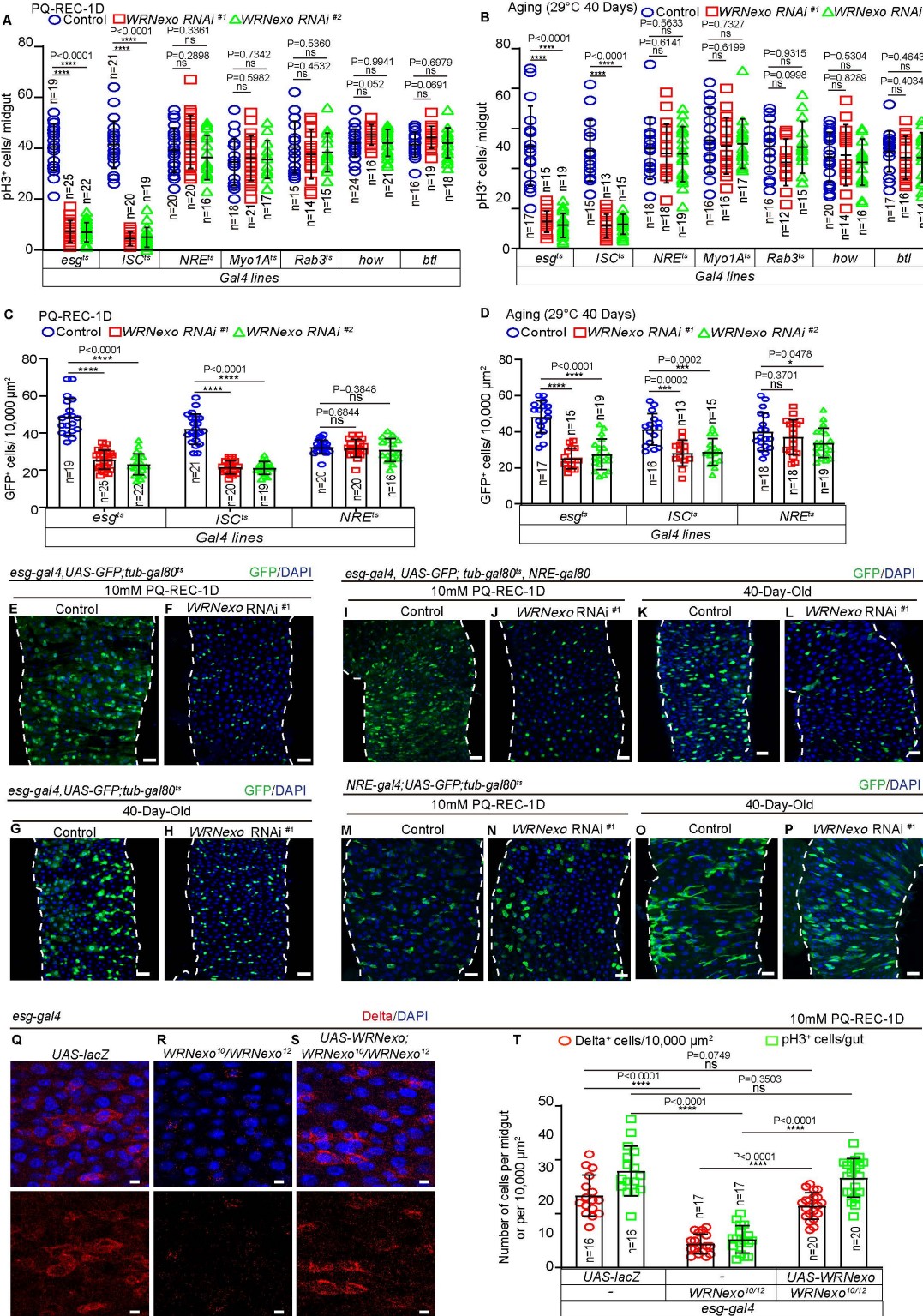

**Fig 3.. WRNexo functions cell-autonomously in ISCs to promote ISC proliferation. (A)** Quantification of pH3+ cells in PQ-REC-1D-treated midguts with control (type-specific Gal4 lines only) and *WRNexo*-depleted flies using type-specific *GAL4*-driven *RNAi*, as indicated. n represents the number of midguts analyzed. **(B)** Quantification of pH3+ cells in aged (40 days) midguts from control (type-specific Gal4 lines only) and *WRNexo*-depleted flies

using type-specific *GAL4*-driven *RNAi*, as indicated. n represents the number of midguts analyzed. **(C)** Quantification of esg-GFP$^+$ cells in PQ-REC-1D midguts with control (type-specific Gal4 lines only) and *WRNexo* depleted by different cell type-specific *GAL4*-driven *RNAi*, as indicated. ROI size was 10,000 μm$^2$, *n* is as indicated. **(D)** Quantification of esg-GFP$^+$ cells of aged (40 days) midguts with control (type-specific Gal4 lines only) and *WRNexo* depleted by different cell type-specific *GAL4*-driven *RNAi* as indicated. ROI size was 10,000 μm$^2$, *n* is as indicated. **(E, F)** Immunofluorescence images of PQ-REC-1D-treated midguts from flies carrying *esg^ts*-*GAL4*-driven *UAS-GFP* (E, control) or *WRNexo RNAi* (F). **(G, H)** Immunofluorescence images of aged midguts from flies carrying *esg^ts*-*GAL4*-driven *UAS-GFP* (G, control) or *WRNexo RNAi* (H). **(I, J)** ISC-specific *GAL4-UAS* system (*ISC^ts*) in control (*ISC^ts*-*GAL4*-driven *UAS-GFP*) (I) and *WRNexo RNAi* flies (J) following PQ-REC-1D treatment. **(K, L)** Immunofluorescence images of aged midguts from flies carrying *ISC^ts*-*GAL4*-driven *UAS-GFP* (K, control) and *WRNexo RNAi* flies (L). **(M, N)** EB-specific *GAL4-UAS* system (*NRE^ts*) in control (*NRE^ts*-*GAL4*-driven *UAS-GFP*) (M) and *WRNexo RNAi* flies (N) following PQ-REC-1D treatment. **(O, P)** Immunofluorescence images of aged midguts from *NRE^ts*-*GAL4*-driven *UAS-GFP* (O, control) and *WRNexo RNAi* flies (P). **(Q–S)** Immunofluorescence images of midguts stained for Delta in *UAS-LacZ* (Q, control), *WRNexo-null* (R), and combined *WRNexo*-overexpressing with *WRNexo-null* (S) flies following PQ-REC-1D treatment. Single-channel images of midguts with Delta immunostaining are shown in the lower panels. **(T)** Quantification of pH3$^+$ cells and Delta$^+$ cells per ROI for midguts of experiments in (Q–S). *n* is as indicated. DAPI-stained nuclei (blue). Scale bars represent 25 μm in E–P and 5 μm in Q–S. Error bars represent SD. Student *t* tests, \**p* < 0.05, \*\**p* < 0.01, \*\*\**p* < 0.001, \*\*\*\**p* < 0.0001, and NS (non-significant) represents *p* > 0.05. See also S3 Table. Underlying data and statistical analysis in S3 Data.

*RNAi* lines showed significantly lower *WRNexo* expression levels compared to wild-type controls (S2O Fig). To further investigate the role of *WRNexo* in ISCs or EBs for regulating ISC proliferation, we overexpressed *WRNexo* using *ISC^ts* (*esg-GAL4, UAS-GFP; tub-Gal80^ts, NRE-Gal80*) or *NRE^ts*(*NRE-GAL4, UAS-GFP, tub-Gal80^ts*) drivers [51], respectively. We found that *WRNexo* overexpression in ISCs, but not in EBs, significantly increased the ISC proliferation rate, as indicated by DI staining (Fig 2Y–2A′). Altogether, these data indicated that WRNexo is essential for the activation of ISC proliferation.

## WRNexo functions cell-autonomously in ISCs to promote ISC proliferation

The endogenous WRNexo reporter line *WRNexo-mCherry* indicated that WRNexo is specifically expressed in ISCs and EBs of young *Drosophila* (Fig 1R–1U). To investigate whether WRNexo functions cell-autonomously in ISCs and EBs or non-cell-autonomously in other intestinal cells to promote ISC proliferation, we used cell type-specific RNAi knockdown of *WRNexo*. We found that *WRNexo* depletion specifically in ISCs but not in EBs, ECs, EEs, muscle cells, or tracheal cells led to ISC proliferation defects. These defects were comparable to those observed in *WRNexo-null* midguts after injury. This was evident from the absence of increases in pH3$^+$ mitotic cells (Fig 3A and 3B) and *esg*$^+$ cells (Fig 3C–3P) in injured or aged midguts.

Importantly, expression of *WRNexo* cDNA using the *esg-Gal4* driver fully restored ISC proliferation in *WRNexo*-null midguts after injury (Figs 3Q–3T, and S3A–S3C). These results demonstrate that WRNexo functions cell-autonomously in ISCs to promote ISC proliferation, ensuring normal gut epithelial homeostasis and regeneration following injury.

## WRNexo regulates ISC proliferation by regulating the activation of UPR$^{ER}$

Since the *WRNexo-null* midguts failed to exhibit ISC proliferation after injury, we aimed to identify the mechanism by which WRNexo promotes ISC proliferation during *Drosophila* midgut regeneration, RNA-seq was performed on dissected midguts from *WRNexo-null* and wild-type flies before and after BLM-induced injury (see S1 and S2 Tables).

RNA-seq analyses revealed that genes involved in proteostasis, including *Hsp67Ba*, *Hsp67Bc*, *Hsp68*, and *Hsp70* family genes, were significantly upregulated in *WRNexo-null* midguts compared to wild-type midguts following BLM-induced injury (Fig 4A and S2 Table). However, no significant differences in these genes were observed between *WRNexo-null* and wild-type midguts before injury (Fig 4A and S1 Table). RT-qPCR validation of selected proteostasis-related genes (*Hsp70Ba*, *Hsp70Bb*, *Hsp68*, *Hsp67Ba*, and *Hsp67Bc*) confirmed expression patterns consistent with those observed in RNA-seq results (S4A and S4B Fig). Gene Ontology analysis of differentially expressed genes that were only upregulated or downregulated in the *WRNexo-null* midguts after BLM-induced injury revealed significant enrichment for genes involved

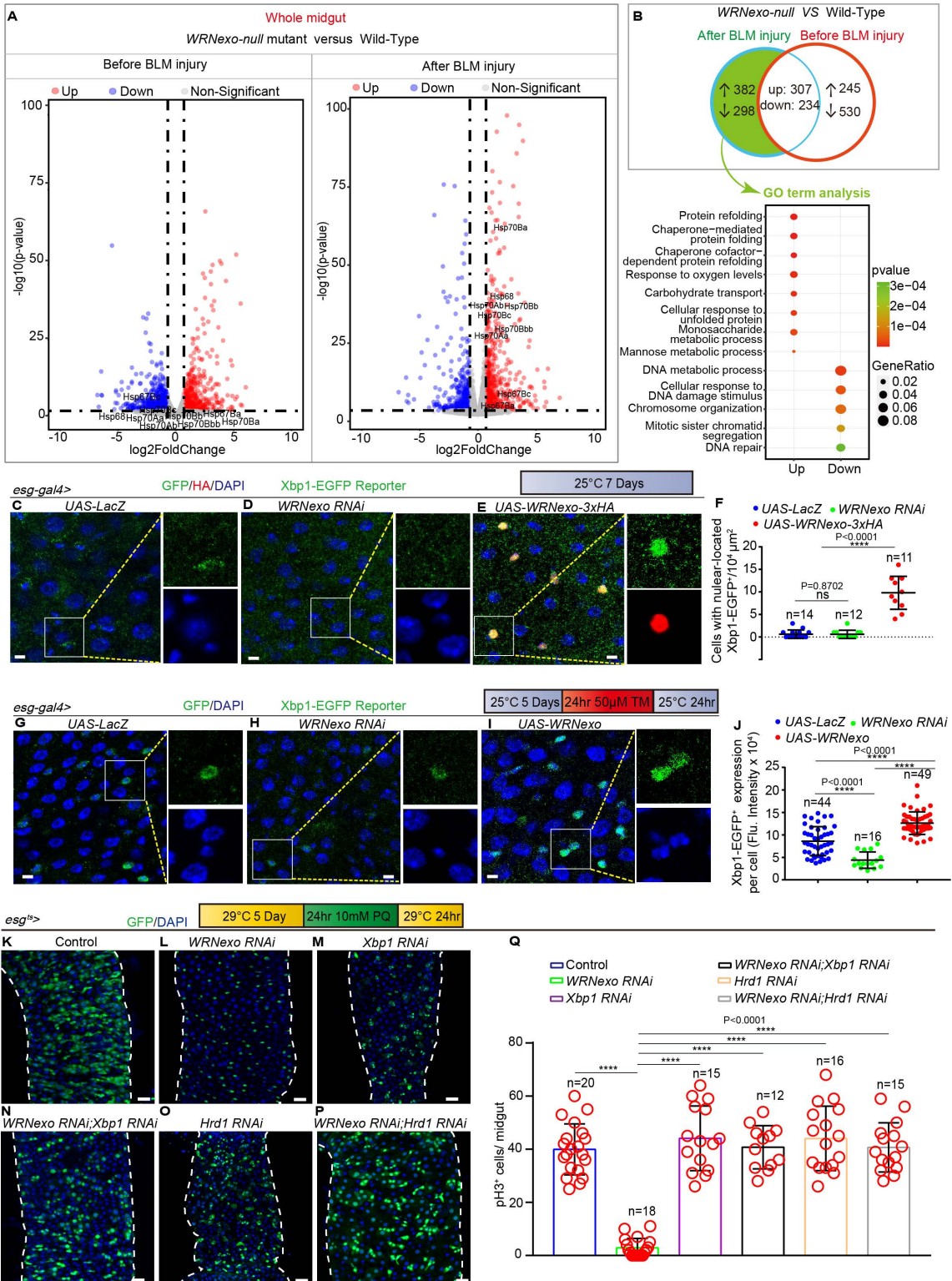

**Fig 4. WRNexo regulates ISC proliferation by regulating the activation of UPR^ER. (A)** Volcano plots of differentially expressed genes in a pair-wise comparison of *WRNexo-null* flies and wild-type (WT) flies before and after BLM-REC-1D treatment. **(B)** Gene Ontology (GO) terms in the biological process (BP) category for differentially expressed genes. Upper panel: Venn diagram of overlap between differentially expressed gene sets from WT

flies and *WRNexo-null* flies before and after BLM-REC-1D treatment. Lower panel: GO term analysis of up- or down-regulated genes limited to the green region in the upper panel. Both the adjusted *p*-value and gene ratio denote the significance of enrichment. **(C–E)** Immunofluorescence images of the midguts of flies carrying the *esg-gal4>UAS-Xbp1-EGFP* reporter system, detecting ER stress through the nuclear translocation of Xbp1-EGFP in *UAS-LacZ* (C, control), *WRNexo RNAi* (D), and *WRNexo* overexpressing flies (E). **(F)** Quantification of the number of Xbp1-EGFP+ cells per ROI in midguts from (C–E). ROI size was 10,000 µm², *n* is as indicated. **(G–I)** Immunofluorescence images of the midguts of flies carrying the *esg-gal4>UAS-Xbp1-EGFP* reporter system, detecting ER stress through the nuclear translocation of Xbp1-EGFP in *UAS-LacZ* (G, control), *WRNexo RNAi* (H), and *WRNexo* overexpressing flies (I). Flies were treated with tunicamycin induction for 1 day and then recovery on regular food for 1 day (TM-REC-1D). **(J)** Quantification of the fluorescence intensity of Xbp1-EGFP+ cell from midguts of experiments in (G–I). Each dot represents a cell, *n* is as indicated. **(K–P)** Immunofluorescence images of midguts from flies carrying *esg^ts-GAL4*-driven *UAS-LacZ* (control, K), *WRNexo RNAi* (L), *Xbp1 RNAi* (M), *WRNexo RNAi* with *Xbp1 RNAi* (N), *Hrd1 RNAi* (O), and *WRNexo RNAi* with *Hrd1 RNAi* (P) under PQ-REC-1D treatment. **(Q)** Quantification of pH3+ cells per whole midgut from experiments in (K–P). Each red circle represents a midgut, *n* is as indicated. DAPI-stained nuclei (blue). Scale bars represent 5 µm in C–E, G–I, 25 µm in K–P. Error bars represent SD. Student *t* tests, *\*p < 0.05, \*\*p < 0.01, \*\*\*p < 0.001, \*\*\*\*p < 0.0001*, and NS (non-significant) represents *p >* 0.05. See also S4 Fig and S1–S3 Tables. Underlying data and statistical analysis in S4 Data.

in protein refolding and the cellular response to unfolded proteins (Fig 4B and S2 Table). The UPR^ER is a cellular stress response triggered by ER stress, which occurs when the protein-folding capacity of the ER is overwhelmed, leading to the accumulation of unfolded or misfolded proteins [52,53]. Upon activation, UPR target genes alleviate ER stress by inducing stress-responsive chaperones [54,55]. The upregulation of multiple proteostasis-related genes in *WRNexo-null* midguts after injury suggests a potential involvement of WRNexo in the regulation of UPR^ER. We hypothesize that WRNexo may regulate UPR^ER activity by suppressing specific proteostasis-related genes.

IRE1/XBP1 is one of the major branches of UPR^ER signaling pathways in *Drosophila* [36]. Moreover, previous studies have shown that the IRE1/XBP1 signaling pathway is involved in metabolic adaptation and life span extension under dietary restriction (DR) in *Drosophila* [56]. Notably, our RNAseq analysis of *WRNexo-null* midguts revealed significant alterations in genes associated with metabolic processes (S1 and S2 Tables). One such gene, *sugarbabe*, a key mediator of IRE1/XBP1-regulated lipogenesis [56], displayed significant variation in *WRNexo-null* midguts (S1 and S2 Tables). Collectively, these findings suggest that WRNexo and UPR^ER-related genes may collaboratively regulate multiple cellular processes, including metabolic adaptation and ER stress response.

The spliced X-box binding protein 1 (Xbp1^s) is a key transcription factor that responds to ER stress and induces genes associated with the UPR, thereby alleviating protein overload in the ER across various *Drosophila* tissues [57–59]. The XBP1-EGFP reporter line is widely used to indicate Xbp1 splicing, with the nuclear translocation of Xbp1^s serving as a visual marker of this splicing and ER-stress activation [36,60,61]. Under normal conditions, Xbp1-EGFP signals were barely detectable in *esg+* cells of *Drosophila*, and no activation of Xbp1 splicing was observed (Figs 4C and S4C). When ER stress is activated, it triggers Xbp1 splicing, leading to the nuclear translocation of Xbp1-EGFP (Fig 4G). Our experiments revealed *WRNexo* overexpression significantly promoted the nuclear translocation of Xbp1-EGFP (Fig 4E and 4F), while *WRNexo* depletion resulted in undetectable nuclear Xbp1-EGFP levels (Fig 4D and 4F). Tunicamycin (TM), a nucleoside antibiotic that inhibits protein glycosylation, is widely used as an ER-stress inducer [62–64]. In our experiments, *WRNexo* overexpression significantly enhanced nuclear Xbp1 signals in ISCs under TM treatment (Fig 4G, 4I, and 4J), whereas *WRNexo* depletion suppressed TM-induced Xbp1 nuclear translocation in ISCs (Fig 4G, 4H, and 4J). Collectively, these findings suggest that *WRNexo* plays a critical role in the ER stress response. To further confirm that WRNexo regulates ISC proliferation by modulating UPR^ER signaling, we increased ER stress by suppressing key UPR regulator genes, Xbp1 and Hrd1. Specifically, Hrd1 encodes a transmembrane ER protein that facilitates the degradation of misfolded peptides [65,66]. Notably, inhibition of Xbp1 through esg-Gal4-driven RNAi restored ISC proliferation in the injured gut epithelium caused by WRNexo deficiency (Fig 4K–4N, and 4Q). Similarly, suppression of Hrd1 also restored ISC proliferation in the injured gut epithelium following WRNexo RNAi (Fig 4K, 4L, and 4O–4Q). These findings illustrate that WRNexo regulates ER stress upstream to modulate ISC proliferation through UPR^ER signaling.

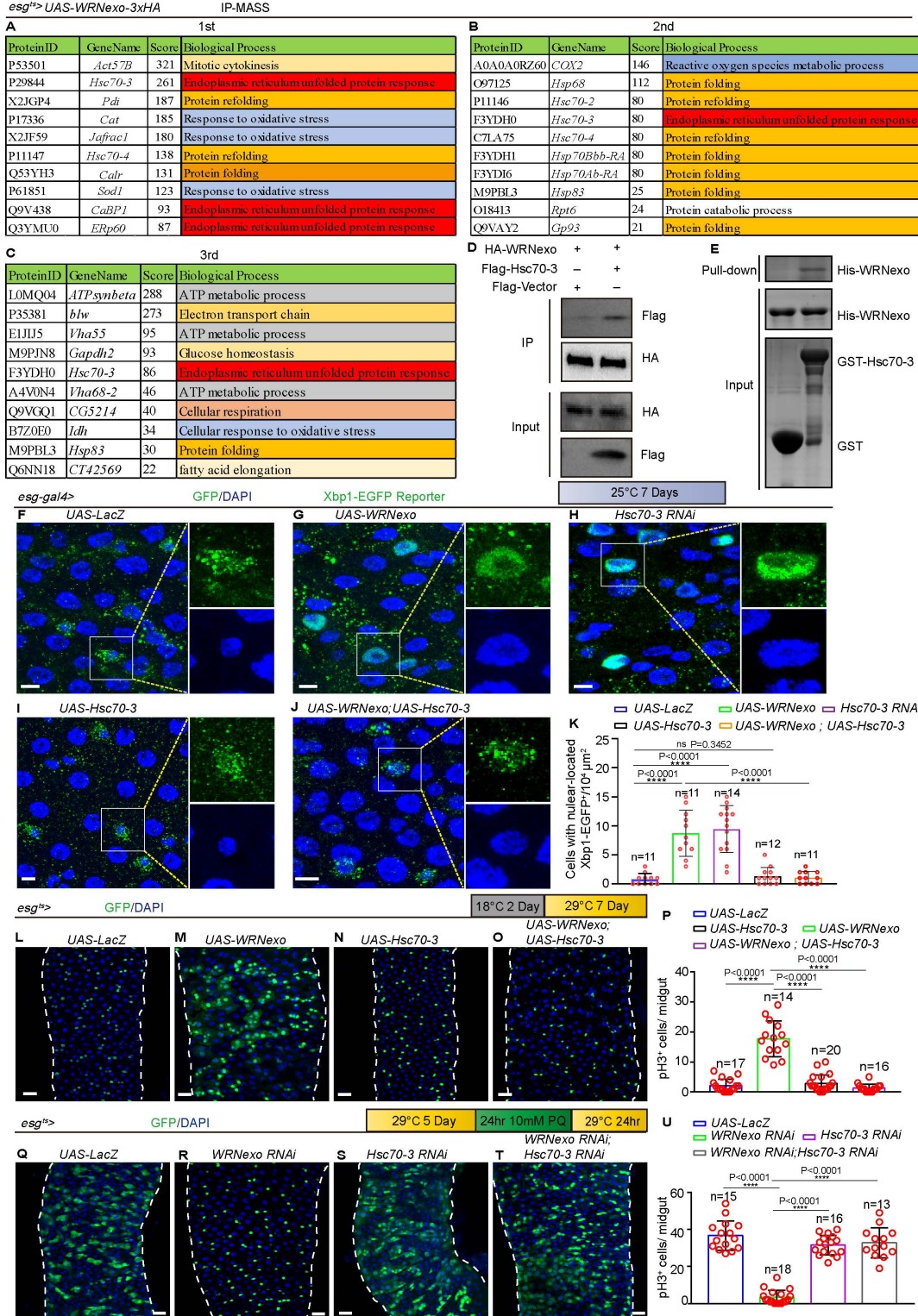

**Fig 5. WRNexo promotes ISC proliferation by the inhibition of Hsc70-3. (A–C)** Partial high score proteins (nuclear protein extraction) in mass spectrometry (MS) analysis of midgut isolated from flies carrying *esg^{ts}-GAL4*-driven *UAS-WRNexo-3xHA*. The Biological Process was analyzed by the DAVID database. Three independent experiments as indicated. **(D)** Co-immunoprecipitation studies were carried out with lysates prepared from 293T cells (a

human embryonic kidney cell line). For the control, 293T cells were co-transfected with Flag-Vector and HA-WRNexo. Simultaneously, 293T cells were co-transfected with Flag-Hsc70-3 and HA-WRNexo. **(E)** Purified His-*Drosophila*-WRNexo was incubated with purified GST or GST-*Drosophila*-Hsc70-3 at 4 °C followed by GST pulldown. Western blotting indicated WRNexo interacted with Hsc70-3. **(F–J)** Immunofluorescence images of the midguts of flies carrying the *esg-gal4>UAS-Xbp1-EGFP* reporter system, detecting ER stress through the nuclear translocation of Xbp1-EGFP in *UAS-LacZ* (F, control), *UAS-WRNexo* (G), *Hsc70-3 RNAi* (H), *UAS-Hsc70-3* (I), and both overexpression of *WRNexo* and *Hsc70-3* (J). **(K)** Quantification of the number of Xbp1-EGFP+ cells per ROI in midguts from (F–J). ROI size was 10,000 µm², *n* is as indicated. **(L–O)** Immunofluorescence images of the midgut of flies carrying *esg$^{ts}$-GAL4*-driven *UAS-LacZ* (L, control), *UAS-WRNexo* (M), *UAS-Hsc70-3* (N), and combined *UAS-WRNexo* with *UAS-Hsc70-3* (O). **(P)** Quantification of pH3+ cells from whole midguts of experiment in (L–O). Each red circle represents a midgut, *n* is as indicated. **(Q–T)** Immunofluorescence images of the midgut of flies carrying *esg$^{ts}$-GAL4*-driven *UAS-LacZ* (Q, control), *WRNexo RNAi* (R), *Hsc70-3 RNAi* (S), and combined *WRNexo RNAi* with *Hsc70-3 RNAi* (T) with PQ-REC-1D treatment. **(U)** Quantification of pH3+ cells from whole midguts of experiment in (Q–T). Each red circle represents a midgut, *n* is as indicated. DAPI-stained nuclei (blue). Scale bars represent 5 µm in F–J, 25 µm in L–O, and Q–T. Error bars represent SD. Student *t* tests, *$p < 0.05$, **$p < 0.01$, ***$p < 0.001$, ****$p < 0.0001$, and NS (non-significant) represents $p > 0.05$. See also S4 Fig and S3–S6 Tables. Underlying data and statistical analysis in S5 Data.

## WRNexo promotes ISC proliferation by the inhibition of Hsc70-3

Our experiments indicate that WRNexo regulates ISC proliferation through UPR$^{ER}$ signaling (Fig 4K–4Q), though the specific mechanism remains unclear. RNA-seq analysis suggests that WRNexo may inhibit proteostasis-related genes, including stress-responsive chaperones (Fig 4A and 4B). We hypothesize that WRNexo modulates UPR$^{ER}$ activity by selectively repressing certain UPR$^{ER}$-related genes. To confirm this hypothesis, we conducted nuclear protein extraction followed by immunoprecipitation (IP) and mass spectrometry (MS) (IP-MS) analyses to investigate the involved protein complexes. By analyzing the three independent experiments on midguts isolated from *esg-Gal4*-driven *WRNexo-HA* overexpression *Drosophila* lines, we identified several high-scoring proteins associated with protein folding, ER unfolded protein response, and ROS metabolic process (Fig 5A–5C and S4–S6 Tables). Surprisingly, among these potential WRNexo-binding proteins, we consistently identified heat shock protein 70 cognate 3 (Hsc70-3/Bip)-an ER-resident chaperone crucial for protein folding and induced by Xbp1$^{s}$ [43,67]-in all independent IP-MS experiments (Fig 5A–5C and S4–S6 Tables). Furthermore, co-immunoprecipitation (Co-IP) analyses confirmed that WRNexo directly binds to Hsc70-3/Bip (Fig 5D) in *Drosophila*.

We acknowledged that Co-IP alone may not conclusively establish a direct interaction between the two proteins. Therefore, we undertook additional experiments, including using AlphaFold to predict their interaction and performing a pull-down assay to confirm it. AlphaFold predictions strongly suggest that *Drosophila* WRNexo (127aa-357aa) directly interacts with *Drosophila* Hsc70-3/Bip protein (S4D Fig). Furthermore, the pull-down assays confirmed the direct interaction between *Drosophila* WRNexo (127aa-357aa) (the WRNexo ortholog in human) [38] and *Drosophila* Hsc70-3/Bip protein (Fig 5E). These data indicate that WRNexo interacts directly with Hsc70-3/Bip in ISCs.

Hsc70-3/Bip (as a key ER stress-responsive regulator) is a direct downstream target of the Xbp1 transcription factor in the regulation of ISC proliferation during *Drosophila* aging [36,57]. Similar to *WRNexo* overexpression, it has been reported that depleting *Hsc70-3* in *esg*+ cells activates ISC proliferation in *Drosophila* midguts [36]. A previous study confirmed that excessive activation of UPR$^{ER}$ signaling can be induced either by misexpressing mutant proteins or by knocking down the ER chaperone Hsc70-3/Bip [68–70]. Therefore, we hypothesized that the overexpressed WRNexo may inhibit Hsc70-3 function by interacting with it.

To investigate whether WRNexo affects UPR$^{ER}$ signaling and ISC proliferation through its interaction with Hsc70-3, we conducted genetic experiments. Hsc70-3/Bip knockdown mimicked the effects of WRNexo overexpression, enhancing UPR$^{ER}$ signaling as indicated by the nuclear localization of XBP1-EGFP (Fig 5F–5H, and 5K). In contrast, co-expression of *Hsc70-3/Bip* and *WRNexo* attenuated the nuclear localization of XBP1-EGFP, suggesting that WRNexo modulates UPR$^{ER}$ signaling through its interaction with Hsc70-3/Bip (Fig 5F–5G and 5I–5K). To further confirm the role of *WRNexo* and *Hsc70-3* in ISC proliferation, we examined their genetic interaction. As expected, overexpression of *Hsc70-3* rescued the *esg*-GFP+ and pH3+ cell accumulation in *WRNexo*-overexpressing midguts (Fig 5L–5P). Additionally, deletion of *Hsc70-3*

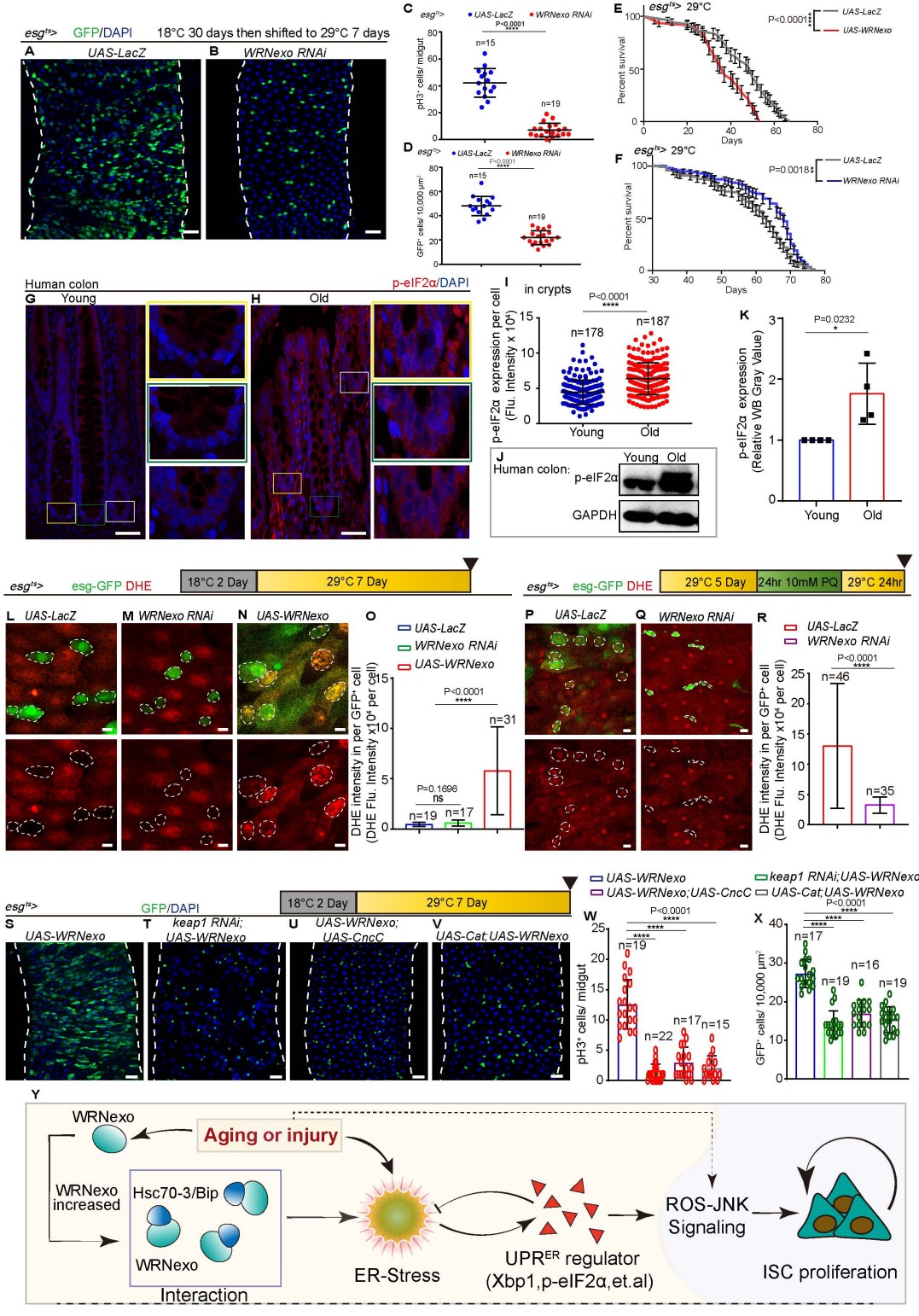

**Fig 6. Age-related upregulation of WRNexo induces gut hyperplasia through ROS signaling. (A, B)** Immunofluorescence images of the midguts of flies carrying *esg^ts-GAL4*-driven *UAS-LacZ* (control, A) and *WRNexo RNAi* (B). Flies were raised on regular food and at a repressive temperature (18 °C) for 30 days after eclosion, then shifted to a permissive temperature (29 °C) for 7 days with normal conditions before dissection. **(C)** Quantification of

pH3+ cells from whole midguts of experiments in (A, B). Each dot represents a midgut, *n* is as indicated. **(D)** Quantification of *esg*-GFP+ cells in the ROI from midguts of experiments in (A, B). ROI size 10,000 µm², *n* is as indicated. **(E)** Survival rate of flies carrying *esg*ts-GAL4-driven *UAS-WRNexo* raised on a permissive temperature (29 °C). Flies carrying *esg*ts-GAL4-driven *UAS-LacZ* served as controls. **(F)** Survival rate of flies carrying *esg*ts-GAL4-driven *UAS-LacZ* (control) and *UAS-WRNexo*. raised on a permissive temperature (29 °C). Flies were raised on regular food and at a repressive temperature (18 °C) for 30 days after eclosion, then shifted to a permissive temperature (29 °C). **(G, H)** Immunofluorescence of p-eIF2α staining in human colons. Normal young para-carcinoma colons (as Young, the mean age was 29 years, *n* = 4, G), normal old para-carcinoma colons from patients with colon cancer (as Old, the mean age was 67 years, *n* = 7, H). Cells were stained by p-eIF2α (red). The boxed areas in (G, H) are enlarged to the right. **(I)** Quantification of the fluorescence intensity of p-eIF2α in experiments (G, H). Each dot represents one cell of boxed regions from crypts. Colon samples were from four young patients and seven old patients. **(J)** Western blotting results indicate an increase in p-eIF2α expression in the colons upon aging. Loading controls, GAPDH. Colon samples were from four young patients and seven old patients. **(K)** Quantification of the relative band intensity of p-eIF2α as shown in experiments (J). **(L–N)** Dihydro-ethidium (DHE) staining of live intestines from flies carrying *esg*ts-GAL4-driven *UAS-LacZ* (L, control), *WRNexo RNAi* (M) or *UAS-WRNexo* (N). DHE single channel is separated in the lower panel. Dashed circles indicate *esg*+ cells. **(O)** Quantification of the fluorescence intensity of DHE per *esg*-GFP+ cell from experiments (L–N). *n* is as indicated. **(P, Q)** DHE staining of live intestines carrying *esg*ts-GAL4-driven *UAS- LacZ* (P, control) or *WRNexo RNAi* (Q) with PQ-REC-1D treatment. DHE single channel is separated in the lower panel. The dashed circle indicates *esg*-GFP+ cells. **(R)** Quantification of the fluorescence intensity of DHE per *esg*-GFP+ cell from experiments (P, Q). *n* is as indicated. **(S–V)** Repression of *Keap1* (*Keap1 RNAi*) (T), overexpression of *CncC* (U), or overexpression of *Cat* (V), all of which could limit ISC proliferation under *WRNexo* overexpression background (S). **(W, X)** Quantification of pH3+ cells from whole midguts (W) and *esg*-GFP+ cells in a 10,000 µm² area from midguts (X) of experiments in (S–V). *n* is as indicated. **(Y)** Schematic summary of the mechanism underlying ISC proliferation caused by WRNexo upregulation in ISCs during aging or regeneration. DAPI-stained nuclei (blue). Scale bars represent 5 µm in L–N and P–Q, 25 µm in A, B, and S–V, 50 µm in G and H. Error bars represent SD. Student *t* tests, *p < 0.05, **p < 0.01, ***p < 0.001, ****p < 0.0001, and NS (non-significant) represents *p* > 0.05. See also S5 and S6 Figs and S3 Table. Underlying data and statistical analysis in S6 Data.

restored ISC proliferation in *WRNexo-null* midguts following injury (Fig 5Q–5U). These data suggest that WRNexo maintains ISC-mediated gut homeostasis by modulating the UPR^ER signaling through the inhibition of Hsc70-3.

## Age-associated WRNexo elevation promotes gut hyperplasia via oxidative stress and JNK signaling

Since *WRNexo* expression increases in ISCs with aging (Figs 1N–1B′ and S1C–S1J), and the overexpression of *WRNexo* in young ISCs results in a phenotype resembling gut hyperplasia in old *Drosophila* (Fig 2Y and 2A′), we predicted that the age-associated increase in *WRNexo* expression may contribute to age-related gut hyperplasia. To test this hypothesis, we used *esg-Gal4*-driven *WRNexo RNAi* to inhibit *WRNexo* expression in middle-age (30-day-old) *Drosophila*. Depleting *WRNexo* reduced age-associated ISC over-proliferation and gut hyperplasia, as indicated by the decreases in *esg*+ cells and pH3+ cells in the midguts of aged *Drosophila* (Fig 6A–6D). Furthermore, *WRNexo* overexpression in young ISCs significantly shortens *Drosophila* life span (Fig 6E). In contrast, reducing *WRNexo* expression in middle-aged flies (30 days old, maintained at 18°C until induction) markedly extended their life span compared to controls (Fig 6F).

To further investigate whether the increase of WRN expression in the crypts of aged colons regulates ISC behavior by the modulation of the UPR^ER, we analyzed colon tissues from elderly individuals. We observed significantly higher levels of phosphorylated eukaryotic initiation factor 2 alpha (p-eIF2α, which is widely used to marker the activation of UPR^ER) [37] in normal colons from elderly individuals compared to those of young controls (Fig 6G–6I). Western blotting further confirmed that p-eIF2α levels were elevated in elderly colons (Fig 6J and 6K). Furthermore, immunostaining analyses revealed an increase in the expression of p-eIF2α in *Drosophila* midguts during aging (S5A, S5C, and S5E Fig). Notably, overexpression of *WRNexo* in young flies led to an upregulation of p-eIF2α (S5A, S5B, and S5E Fig), while p-eIF2α levels were significantly reduced in aged flies with *WRNexo* knockdown (S5C–S5E Fig). These findings suggest that WRN may play a role in human gut homeostasis and aging through a UPR^ER-mediated mechanism in ISCs.

Prior research has demonstrated that the UPR^ER involves the proliferation of ISC in *Drosophila* through oxidative stress signaling [36]. Besides, WRNexo may involve the process of response to oxidative stress (Fig 5A–5C, and S4–S6 Tables). Consequently, we conducted additional investigations to determine whether WRNexo also stimulates ROS activation in ISCs. To ascertain if WRNexo elevates ROS levels in midguts, we employed dihydroethidium (DHE) for monitoring of endogenous ROS levels in vivo [71]. Knocking down *WRNexo* in *esg*+ cells under normal conditions did not significantly change the level of ROS in *esg*+ cells when compared to levels in the control (Fig 6L, 6M, and 6O). ROS levels were higher

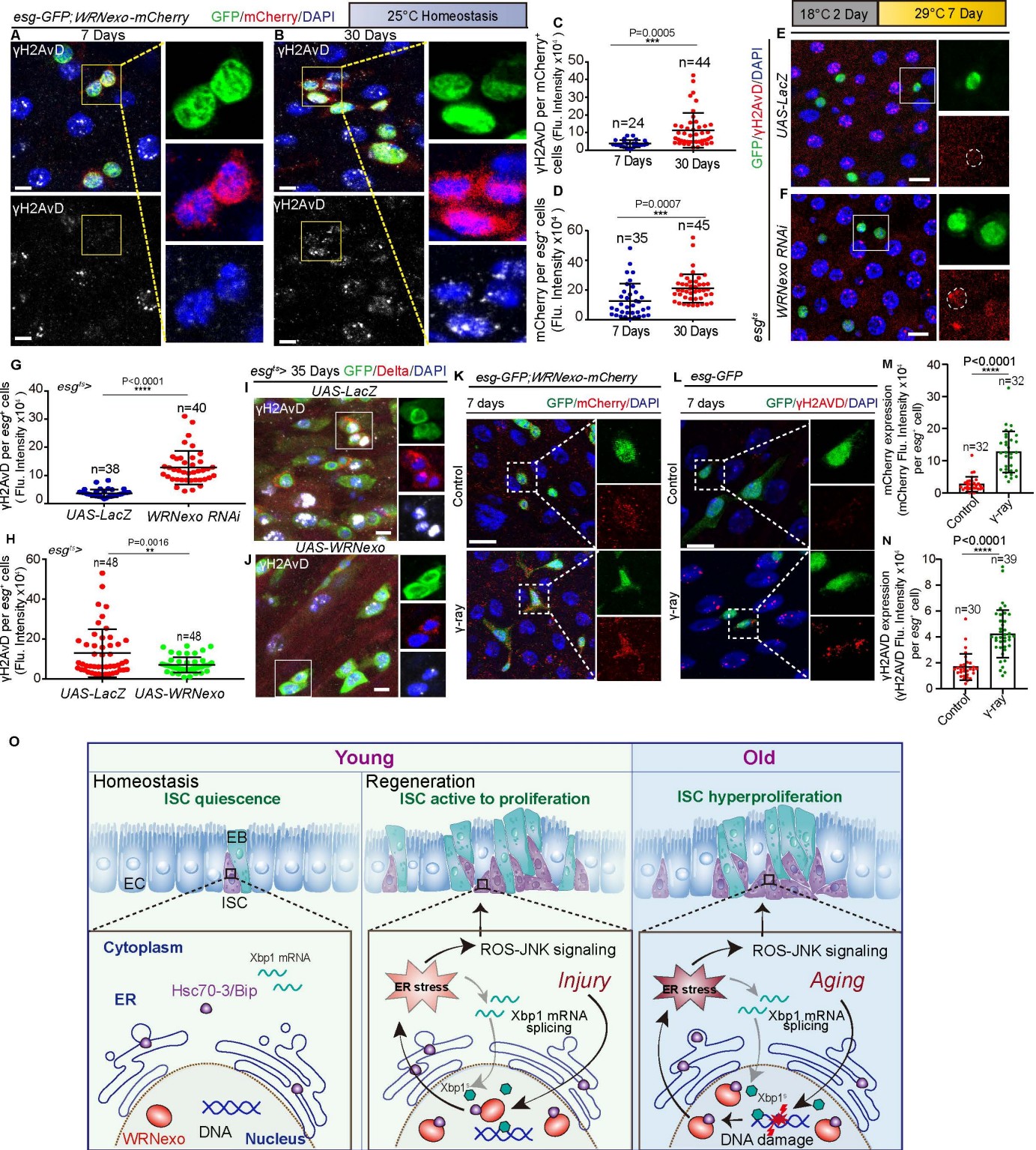

**Fig 7. Age-associated WRNexo elevation is induced by the accumulation of DNA double-strand breaks in ISCs. (A, B)** Immunofluorescence images of young (A) and old (B) midguts of flies carrying *esg-GFP* and *WRNexo-mCherry* endogenous reporter. γH2AvD single channel image is shown in the lower panel. The boxed areas in immunofluorescence images of GFP, mCherry, and γH2AvD staining are enlarged on the right. **(C)** Quantification

of the fluorescence intensity of γH2AvD in per WRNexo-mCherry$^+$ cell of young (A) and old (B) midguts. Each dot represents a WRNexo-mCherry$^+$ cell, *n* is as indicated. **(D)** Quantification of the fluorescence intensity of WRNexo-mCherry per *esg*-GFP$^+$ cell of young (A) and old (B) midguts. Each dot represents an *esg*-GFP$^+$ cells, *n* is as indicated. **(E, F)** Representative images of the young midgut of flies carrying *esg$^{ts}$-GAL4*-driven *UAS-LacZ* (control, E) and *WRNexo RNAi* (F) under permissive temperature, stained with γH2AvD and GFP. Dashed line indicates *esg*-GFP$^+$ cells. The boxed areas in immunofluorescence images of γH2AvD and GFP staining are enlarged on the right. **(G)** Quantification of the fluorescence intensity of γH2AvD per *esg*-GFP$^+$ cell of midguts from control (E) and depletion of *WRNexo* in *esg$^+$* cells (F). Each dot represents an esg-GFP$^+$ cell, *n* is as indicated. **(H)** Quantification of the fluorescence intensity of γH2AvD per *esg*-GFP$^+$ cell of midguts from control (I) and *WRNexo*-overexpressing *esg$^+$* cells (J). Each dot represents an *esg*-GFP$^+$ cell, *n* is as indicated. **(I, J)** Representative images of aged (35 days) midguts of flies carrying *esgts-GAL4*-driven *UAS-LacZ* (control, I) and *WRNexo* overexpression vectors (J), stained with γH2AvD and GFP. The boxed areas in immunofluorescence images of GFP, Delta, and γH2AvD staining are enlarged on the right. **(K)** Immunofluorescence images of the control (non-γ-irradiation treatment, upper panel) and γ-irradiation (lower panel) midguts from flies carrying *esg-GFP* and *WRNexo-mCherry* endogenous reporters. The boxed areas in immunofluorescence images of GFP and mCherry staining are enlarged on the right. **(L)** Immunofluorescence images of the control (non-γ-irradiation treatment, upper panel) and γ-irradiation (lower panel) midguts from flies carrying *esg-GFP* endogenous reporters. The boxed areas in immunofluorescence images of GFP and γH2AvD staining are enlarged on the right. **(M)** Quantification of the fluorescence intensity of *WRNexo-mCherry* midgut-derived per *esg*-GFP$^+$ cell of control (upper panel, K) and γ-irradiation treatment (lower panel, K). Each dot represents an *esg*-GFP$^+$ cell, *n* is as indicated. **(N)** Quantification of the fluorescence intensity of γH2AvD midgut-derived per *esg*-GFP$^+$ cell of control (upper panel, L) and γ-irradiation treatment (lower panel, L). Each dot represents an *esg*-GFP$^+$ cell, *n* is as indicated. **(O)** In *Drosophila* ISCs, WRNexo upregulation and Hsc70-3/Bip translocation to nuclei occur with injury/aging. WRNexo interaction inhibits Hsc70-3/Bip, increasing ER stress, which promotes ISC proliferation via ROS-JNK signaling. DAPI-stained nuclei (blue). Scale bars represent 5 μm in A, B, K, and L, 10 μm in E, F, I, and J. Error bars represent SDs. Student *t* tests, *$p < 0.05$, **$p < 0.01$, ***$p < 0.001$, ****$p < 0.0001$, and NS (non-significant) represents $p > 0.05$. See also S7 Fig and S3 Table. Underlying data and statistical analysis in S7 Data.

in *esg$^+$* cells with *WRNexo* overexpressed than in the control (Fig 6L, 6N, and 6O). Given that PQ can elevate intracellular ROS levels [21], we measured ROS levels in flies expressing *WRNexo RNAi* under the control of *esg$^{ts}$-Gal4* after PQ-induced injury. ROS levels in *esg$^+$* cells were significantly lower in WRNexo RNAi flies than in controls after PQ treatment (Fig 6P–6R). Furthermore, we used the MitoTimer (MitoTimer encodes a mitochondria-tagged green fluorescent protein and shows a shift to red when the proteins are oxidized) reporter line to detect the cellular redox state [72]. We found that the overexpression of *WRNexo* in *esg$^+$* cells could elevate ROS levels compared to those in controls (S5F–S5H Fig).

Given that previous studies have demonstrated that the regulation of the cellular redox state by *CncC*, *Keap1*, and *Catalase* (*Cat*) (Fig 5A–5C and S4–S6 Tables) [31,34], we investigated the genetic interaction between *WRNexo* and these genes associated with ROS. Our results showed that reduction of cellular ROS levels, achieved by either overexpressing *Cat* or *CncC* or by depleting *Keap1*, could reverse the ISC proliferation induced by *WRNexo* overexpression in *esg$^+$* cells (Fig 6S–6X). The findings indicated that the increase in ISC proliferation induced by *WRNexo* overexpression could be reversed by reducing cellular ROS.

Previous studies have reported that UPR$^{ER}$-related gut dysplasia in *Drosophila* is caused by the activation of JNK signaling in ISCs upon aging [36,57]. Accordingly, we evaluated whether *WRNexo* causes ISC hyperplasia via JNK signaling. As expected, the expression of pJNK was increased with *WRNexo* overexpressing during homeostasis and decreased in aged flies with *WRNexo* depletion (S6A–S6D Fig). Moreover, the expression of a dominant-negative (DN) version of *Basket* (*Bsk*; the *Drosophila* homolog of JNK) reduced ISC proliferation in *Drosophila* with *Hsc70-3 RNAi* following gut injury (S6E–S6H Fig). The expression of constitutively active (CA) *Hemipterous* (*Hep*; which encodes a JNK kinase in *Drosophila*) restored ISC proliferation defects in *WRNexo RNAi* flies after injury (S6I–S6M Fig). Furthermore, we found that gut hyperplasia caused by *WRNexo* overexpression was rescued by expressing a DN version of *Bsk* (S6N–S6Q Fig). These results suggested that JNK activation is necessary for WRNexo to promote ISC proliferation. Altogether, *WRNexo* upregulation in ISCs of aged *Drosophila* causes ISC hyper-proliferation and gut dysplasia by promoting UPR$^{ER}$-mediated ROS and JNK signaling (Fig 6Y).

## Age-associated WRNexo elevation is induced by the accumulation of DNA double-strand breaks in ISCs

In young *Drosophila* midguts, ISCs are predominantly quiescent but are rapidly activated to proliferate and replenish damaged epithelial cells following gut injury [35]. As *Drosophila* age, the proportion of quiescent ISCs gradually decreases, leading to gut hyperplasia characterized by the accumulation of *esg$^+$* and Dl$^+$ cells in the midguts [10,21]. Our study

revealed that injury signals from the gut epithelium induce WRNexo upregulation in ISCs of young *Drosophila* midguts. However, the underlying mechanism driving the age-related increase in WRNexo expression in ISCs remains unclear, warranting further investigation into the regulatory pathways contributing to this phenomenon.

The accumulation of DNA double-strand breaks (DSBs) is a well-defined hallmark of animal aging [5,35]. Interestingly, WRN is a key regulator for the repair of DSBs in cultured human cells [6], and studies have shown that DSB induction leads to WRN accumulation in these cells [73]. In *Drosophila*, WRNexo depletion has been reported to disrupts the genome integrity in wing hairs [38,46] and increase DSBs levels in embryos [39]. Therefore, we hypothesized that the age-related upregulation of WRNexo in ISCs might be linked to the accumulation of DSBs during aging.

To test this hypothesis, we analyzed double-staining confocal images of γH2AvD, a marker for DSBs in *Drosophila*, and endogenous *WRNexo-mCherry* in ISCs of aging flies. Our results revealed that WRNexo upregulation in ISCs and EBs closely correlated with the distribution pattern of DSBs in aged flies (Fig 7A–7D). In addition, a much higher γH$_2$AvD signal was detected in the ISCs and EBs of young flies with *WRNexo*-depleted in *esg*⁺ cells (Fig 7E–7G).

The overexpression of *WRNexo* by *esg-Gal4* significantly reduced the γH$_2$AvD signal in Dl⁺ ISCs of aged *Drosophila* compared to control flies (Fig 7H–7J). To directly induce DSBs, we employed γ-irradiation [74,75] and observed that DSBs indeed promoted the upregulation of WRNexo in *Drosophila* ISCs (Fig 7K–7N). To further investigate the association between DNA damage and WRNexo expression, we manipulated DNA damage repair genes, such as ATM and Chk1 [76]. Specifically, knockdown of ATM or Chk1, which impairs DNA repair capacity, in middle-aged flies led to a significant increase in WRNexo expression in ISCs (S7A–7H Fig). This result was consistent with our observations from γ-Irradiation experiments (Fig 7K–7N). Collectively, these findings suggest that the increase in WRNexo expression in ISCs of aging flies is likely associated with the accumulation of DNA damage in ISCs.

## Discussion

WRN deficiency is a known cause of adult progeria (Werner syndrome), and as such, it has been extensively studied in cultured cells derived from Werner syndrome patients and in model organisms with WRN depletion [6,73,77]. However, its physiological role in normal tissue homeostasis and natural aging remains poorly understood. WRN protein primarily functions as a DNA helicase in cells, which has been implicated in the accumulation of DNA damage—a key factor proposed to drive cellular senescence [78,79]. Consistent with this, cells from Werner syndrome patients display heightened genomic instability and hypersensitivity to DNA damage-inducing agents [80,81]. Interestingly, some studies have reported that inhibitors of WRN helicase activity do not induce hypersensitivity in WRN-deficient cells, and their drug sensitivity profiles instead implicate the exonuclease function of WRN [81,82]. Moreover, homozygous point mutations that inactivate WRN helicase activity have been shown not to result in clinical manifestations of Werner syndrome [83], further highlighting the importance of the exonuclease function. *Drosophila* WRN (WRNexo), which lacks helicase activity and retains only exonuclease function, offers a unique model system to specifically investigate the role of WRN exonuclease activity in aging and tissue homeostasis. This model system may help clarify the distinct contributions of WRN enzymatic functions in aging-related pathologies. While as WRN's helicase activity may contribute to its role in aging in humans, future investigations should focus on mammalian models to determine whether WRN upregulation represents a conserved mechanism underlying stem cell aging and age-related pathologies in humans.

Our study demonstrates that WRNexo in ISCs is essential for gut epithelial repair under the normal homeostatic state but contributes to gut hyperplasia during aging. WRNexo expression is highly induced in ISCs upon gut epithelial damage or in response to DNA damage accumulation in aged animals. The elevated WRNexo interacts with Hsc70-3/Bip, a key ER stress-responsive regulator, and inhibits its activity. This inhibition leads to ER stress, upregulates ROS-JNK signaling, and ultimately drives ISC hyper-proliferation (Fig 7O). This study provides the first evidence that WRNexo plays a dual role: it is critical for stem cell-mediated tissue regeneration but also promotes age-associated tissue dysplasia.

It is essential to emphasize that our study presents evidence that the absence of WRNexo does not trigger the UPR during homeostasis. However, following gut injury, proteostasis-related genes such as *Hsp67Ba*, *Hsp67Bc*, *Hsp68*, *and Hsp70* family become activated in *WRNexo*-deficient conditions. Recent studies have shown that human ER luminal chaperone GRP78 (also known as Bip and encoded by the *HSPA5* gene), an ortholog of *Drosophila* Hsc70-3 [84], is up-regulated and translocated to the nucleus in cancer and stressed cells [85]. Based on this evidence, we hypothesized that the up-regulation and nuclear translocation of GRP78 might explain why Hsc70-3 translocates to the nucleus and interacts with the WRNexo protein (Figs 5A–5C, and 7O and S4–S6 Tables). Earlier investigations have unveiled the coordinated interplay between UPR^ER and ROS in regulating the proliferation of ISCs [36]. In line with these findings, we observed that the absence of WRNexo prevented the elevation of cellular ROS levels upon PQ exposure. This observation aligns with recent research demonstrating that impaired mitochondrial function and metabolic dysfunction are hallmark features of Werner syndrome [86,87]. Additionally, previous studies have highlighted the role of the IRE1/XBP1 signaling pathway module in metabolic adaptation under DR in flies [56]. Transcriptomic analysis of *Drosophila* midgut ECs under DR revealed significant changes in genes related to metabolic processes, such as *GstD9*, *Lip3*, and *CG10383* [56]. Intriguingly, our RNAseq analysis of *WRNexo-null* midguts identified similar alterations in metabolic processes, with notable changes in the expression of these genes (Fig 4B and S1 and S2 Tables). These findings suggest a conserved link between *WRNexo*, metabolic regulation, and gut homeostasis.

Over the past decade, studies of WRN have focused on its role in Werner syndrome, while little is known about its function in natural aging. A prior study demonstrated that WRN levels were significantly lower in primary dental pulp mesenchymal stem cells derived from old individuals compared to young individuals [6]. However, in vivo studies examining WRN expression in ISCs during normal aging are largely absent. The endogenous WRNexo reporter line, *WRNexo-mCherry*, revealed that WRNexo expression was primarily confined to *esg*+ ISCs and EBs, with minimal expression in fully differentiated ECs and EEs in *Drosophila* midguts. These results suggest that WRNexo functions in stem and/or progenitor cells. We further demonstrated that the age-associated increase in WRNexo expression in ISCs is a response to accumulation of DNA damage. WRNexo upregulation was shown to be essential for maintaining gut homeostasis and facilitating acute regenerative responses. However, chronic overexpression of WRNexo in ISCs during aging leads to detrimental effects, contributing to gut dysplasia in older flies. These findings provide novel insights into how WRN expression dynamics contribute to age-associated tissue dysfunction.

The progressive decline of proliferative homeostasis is a hallmark of aging in the intestinal epithelium, contributing to age-associated diseases such as colorectal cancers. The long-term maintenance of intestinal epithelial homeostasis depends on ISCs. However, due to their longevity and frequent replication, ISCs are more susceptible than terminally differentiated epithelial cells to endogenous and exogenous stressors, including DNA damage, oxidative stress, and ER stress [88]. To counteract these challenges, stem cells have evolved effective repair or response mechanisms, such as DNA damage response (DDR), UPR^ER, and diverse antioxidants [35,89]. Although substantial progress has been made in elucidating the signaling pathways and responses to these stressors, the specific factors driving stem cell aging remain poorly understood. Using *Drosophila* ISCs as a model, our findings suggest that the accumulation of DNA damage is a key contributor to increased WRNexo levels during aging (Fig 7). Aging, however, is a multifaceted process influenced by factors such as elevated ROS production and ER stress, both of which may also play roles in regulating WRNexo expression. Further studies are needed to explore the complex interplay between these factors and their contributions to WRNexo regulation during aging.

An effective DDR is essential for the longevity and high replication potential of stem cells. While DDR impairment has been implicated in tumor formation [90], the potential side effects of its over-activation remain largely unexplored. Recent studies have demonstrated that WRN activation is critical for the survival of cancer cells characterized by MSI [7,8]. Consequently, WRN is now being investigated as a novel DDR target in preclinical and clinical studies [91]. Our findings in the *Drosophila* midgut model suggest that WRNexo over-activation may promote tumorigenesis by inducing ER-stress

and upregulating JNK signaling. This study highlights a potential mechanism by which over-active DDR could exacerbate tumor development through enhanced ER stress and its downstream signaling pathways.

ER stress and the UPR[ER] are well-studied in the context of stem cell aging [35,61]. Impairment of the UPR[ER] results in the accumulation of misfolded proteins in the ER of aged stem cells [35,36]. While numerous studies have focused on the downstream effects and signaling factors of the UPR[ER] [36,37,60], the upstream regulation of the UPR[ER] remains largely unexplored. In our study, we found that WRNexo regulates UPR[ER] activation by interacting with Hsc70-3/Bip. Our results identify a previously unrecognized mechanism underlying alterations in the UPR[ER] in *Drosophila* ISCs during aging. Future studies should focus on elucidating the detailed mechanism through which WRNexo interacts with Hsc70-3/Bip to regulate the transcription of other UPR-related genes in ISCs. This will provide further insight into how UPR[ER] dysregulation contributes to stem cell aging and associated pathologies.

## Materials and methods

### Human samples

Human colon tissues from patients with colorectal carcinoma were provided by Doctor Lishou Xiong of the First Affiliated Hospital of Sun Yat-sen University. The intestinal tissues were collected from 13 patients with peritumoral tissues and 4 patients with colorectal cancer at the First Affiliated Hospital of Sun Yat-sen University (see Table 1 for detailed patient information). The gut samples were checked on the day of surgery and mucosal biopsies were ensured during endoscopy. Patients had pathologically and clinically diagnosed colorectal cancer [92,93].

Colon segments were resected during surgery, and normal and tumor/disease tissues were separated. All participants provided written informed consent for this study. This study protocol complied with the ethical guidelines of the 1975 Declaration of Helsinki principles and was approved by the Human Ethics Committee of Sun Yat-sen University (Approval numbers: [2022] No. 133).

### *Drosophila* breeding and maintenance

The age and rearing conditions of *Drosophila* are described in the text, figures, legends, and Methods. The stocks of *Drosophila* were maintained on standard yeast food (cornmeal 50 g, yeast 18.5 g, sucrose 80 g, glucose 20 g, agar 5 g, and propionic acid 30 mL combined in 1 L of water) at 25 °C, 65% humidity, and on a normal 12-h light/dark daylight cycle, unless otherwise stated.

The temporal and regional gene expression targeting method was used for *Gal4-UAS*-mediated RNAi and overexpression experiments. To repress the *Gal4* system, the crosses were maintained at 18°C when driving temperature-sensitive *Gal4*-mediated RNAi or gene overexpression. During eclosion or at a certain age after eclosion, adults were shifted to 29 °C to turn on the *Gal4* system, which induces RNAi or gene overexpression. Enclosed flies were incubated at 29 °C for the indicated time period, followed by the dissection of midguts for immunostaining or western blotting. Mated females were used for experiments on the *Drosophila* midguts. All fly lines were backcrossed into the *w[1118]* background for six generations, and sibling populations were derived from the crosses. The unmarked fly lines were genotyped by non-lethal

**Table 1. Demographic and clinical characteristics of all participants.**

| Participant characteristics | PT | CRC |
| --- | --- | --- |
| Number of patients | 13 | 4 |
| Age, years (median, range) | 61 (26–71) | 33 (26–37) |
| Gender (male/female) | 8/5 | 3/1 |

PT, peritumoral tissue; CRC, colorectal cancer.

PCR genotyping using adult wings from individual *Drosophila*, as previously described [94,95]. Only flies with the correct genotype were selected for the next round of backcrossing. The final offspring, which had successfully passed the back-crossing process, were used for experimentation.

For the aging-associated experiments, *tub-Gal80ts* transgenic *Drosophila* combined with *esg-Gal4* were grown at a permissive temperature (18 °C) to limit *Gal4* activity. Adult flies were raised at 18 °C for 30 days after eclosion, then transferred to a non-permissive temperature (29 °C) for 7 days until the dissection of midguts for immunostaining analyses.

### *Drosophila* lines

All Drosophila strains used in this study are female and listed below. All genotypes for experimental crosses are provided in S3 Table.

The following *Drosophila* lines and stocks were collected from the Bloomington *Drosophila* Stock Center (BDSC), the Vienna *Drosophila* Resource Center (VDRC), the TsingHua Fly Center (THFC), or the Zurich ORFeome Project (FlyORF).

The following *Drosophila* lines were obtained from the FlyORF: *UAS-CncC* (F000602) and *UAS-Hsc70-3*(F000956). The following *Drosophila* lines were obtained from the BDSC: The *w1118* allele (BDSC3605) was used as the wild-type control. *UAS-LacZ* (BDSC 8529), *UAS-LacZ* (BDSC 3956), *UAS-GFP* (BDSC 4776), *how-Gal4* (BDSC 1767), *UAS-WRNexo RNAi* (BDSC 38297), *UAS-Xbp1-EGFP* (BDSC 60730), *Xbp1 RNAi* (BDSC 36755), *Hrd1 RNAi* (BDSC 50609), *keap1 RNAi* (BDSC 40932), *UAS-Cat* (BDSC 24621), *UAS-hepCA* (BDSC 6406), *Hsc70-3 RNAi* (BDSC 32402), *UAS-BSKDN* (BDSC 6409), and *UAS-Mito Timer* (BDSC 57323). The following *Drosophila* lines were obtained from the VDRC: *UAS-WRNexo RNAi* (VDRC 100227), *UAS-WRNexo RNAi* (VDRC 44595) and *btl-Gal4* (VDRC 109128). The following *Drosophila* lines were obtained from the THFC: *ATM RNAi* (THU5591) and *Chk1 RNAi* (THU2601).

The following *Drosophila* lines were kindly provided as indicated: *esg-GFP* line, *esg-Gal4* line, and *tub-Gal4* line by Dr. Allan Spradling; *esgts-Gal4* line, *MyoIAts-Gal4* line, *ISCts-Gal4* line, *Rab3ts-Gal4* line, *NREts-Gal4* line, and reporter line *NRE-lacZ* by Dr. Benjamin Ohlstein; *FRT82B* and *esgts, UAS-flp, act>CD2>gal4, UAS-GFP* line provided by Dr. Zheng Guo. The following *Drosophila* lines were available in our laboratory: *WRNexo-mcherry*, *WRNexo Mutant WRNexo10*, *WRNexo12*, *WRNexo15*, *UAS-WRNexo-3×HA*, and *UAS-WRNexo*.

### Mouse samples

A total of 80 male C57BL/6J mice (six weeks of age) with a weight range of 18–22 g were obtained from GemPharmatech Co., Ltd (Nanjing, Jiangsu, China). The mice were housed under an automated 12-h light/dark cycle, with a controlled relative humidity of 40%–50% and a temperature of 22 ± 2 °C, and were provided with a standard dry diet and tap water ad libitum. The animals received humane care, and all experimental procedures were performed in accordance with the guidelines for the health and care of experimental animals established by Sichuan University (Chengdu, China). Ethical approval was granted (Permit Number: 202203170).

## Method details

### Generation of *WRNexo* mutant and transgenic lines

Three *WRNexo* null mutants were generated by Cas9-mediated gene knockout, as described previously [96]. The following guide RNAs were used:

single-guide RNA-1-Forward:

5′-GTCGGAAAAGCAAATGAAGTTCCCA-3′

single-guide RNA-1-Reverse:

5′-AAACTGGGAACTTCATTTGCTTTTC-3′

single-guide RNA-2-Forward:

5′-GTCGAGGAGACTCCCAAAGTGGCA-3′

single-guide RNA-2-Reverse:

5′-AAACTGCCACTTTGGGAGTCTCCT-3′

The transgenic *Drosophila* lines *UAS-WRNexo* and *UAS-WRNexo-3×HA* were constructed in-house. In brief, cDNA of *WRNexo* was cloned into the pEntry or pEntry-3×HA vector [97] using the pEASY-Uni Seamless Cloning and Assembly Kit (CU101-02; TransGen Biotech, Beijing, China). Then, the previous product was sub-cloned into the pTW vector by LR recombination. Finally, the vectors were cloned, verified by sequencing, and sent to UniHuaii Corporation (Zhuhai, China) for *Drosophila* embryo injection. The *WRNexo* cDNA primers were as follows:

*WRNexo* L: 5′-TTGCGGCCGCATGGAAAAATATTTAACAAAAATGCCCA-3′;

*WRNexo* R: 5′-ATGGGTAGAGACCCAGAGTCACCTCGTTGATCTTGGT-3′.

Knock-in lines were made in-house. *WRNexo-mCherry* lines and various vectors were verified by sequencing and injected through Fungene Biotechnology (Beijing, China). To construct knock-in *Drosophila* lines, two constructs were generated, one with two sgRNAs and another with the homologous recombination sequence. The two sgRNAs were used to generate specific gene deletion regions. The sgRNA sequence was composed in vitro and sub-cloned into a PMD18T vector to get the U6 promoter. The U6 promoter and sgRNAs were amplified by PCR in the PMD18T vector. Two products with the U6 promoter and sgRNAs were sub-cloned into the PCR8 vector by Golden Gate assembly and recombined into the attB vector by LR recombination to generate the sgRNA construct. To generate the homologous recombinant, the 5′ homology arm (~1 kb), mCherry, and the 3′ homology arm (~1 kb) were inserted into the PASK vector (obtained from the Fungene Biotechnology). The 5′ homology arm and 3′ homology arm were used to repair homologous recombination and mCherry was introduced by homologous recombination. The sgRNAs were designed using http://targetfinder.flycrispr.neuro.brown.edu/ and are listed below.

Target1-*WRNexo*-sgRNA:

Forward: 5′-GTCGAGATCAACGAGGTGACTCTG-3′

Reverse: 5′-AAACCAGAGTCACCTCGTTGATCT-3′

Target2-*WRNexo*-sgRNA:

Forward: 5′-GTCGTCTAATTCCTTTTACCTGTT-3′

Reverse: 5′-AAACAACAGGTAAAAGGAATTAGA-3′

Primers for *mCherry* and *loxP* in the vector:

Forward: 5′-GTGAGCAAGGGC-GAGGAGGAT-3′

Reverse: 5′-ATAACTTCGTATAATGTATGCTATACGAAGTTATCAGCT-TCGCATGGTT-3′

Primers for the vector arm:

5′ Arm:

Forward: 5′-CCTCTTCGCTATTACGCCAGGATGGAGGAAG-AAAATCCGCCCA-3′;

Reverse: 5′-ATCCTCCTCGCCCTTGCTCACACCTCCCAGAGTCACCTCGTTGA-TCTTCGTCAGAAA-3′.

3′ Arm:

Forward: 5′-GCATACATTATACGAAGTTATGCTTTTCTTTGAAG-TTTCAGATATGCGATCTAATT-3′.

Reverse: 5′-GCTATGACCATGATTACGCCACATGAGGCATCCACAACAGCGCA-3′.

## Mosaic analysis with a repressible cell marker

We used *WRNexo* mutant and FLP/FRT-mediated mitotic recombination technique to generate *WRNexo*-null MARCM clones. The *WRNexo¹⁰*, *WRNexo¹²*, and *WRNexo¹⁵* respectively, carrying the *FRT82B* line was obtained as a gift from Dr. Zheng Guo. To generate *UAS-WRNexo* MARCM clones, *FRT82B* was crossed to *UAS-WRNexo* to produce *UAS-WRNexo/cyo; FRT82B/TM6B* flies. Then, these *Drosophila* were crossed to *yw hsFLP::tub-Gal4::UAS-nls GFP/FM7;+/+; tubG80 FRT82B/TM6B* (a gift from Dr. Zheng Guo) was used to obtain *yw hsFLP::tub-Gal4::UAS-nls GFP/+; WRNexo¹⁰ or ¹²::tubG80 FRT82B/ FRT82B*, and *yw hsFLP::tub-Gal4::UAS-nls GFP/+; UAS-WRNexo/+; WRNexo¹⁰ or ¹²::tubG80 FRT82B/ FRT82B* flies. The crosses were kept at 25 °C.

For MARCM clone production, adult *Drosophila* were fed at 25 °C for 2–3 days and heat-shocked at 37 °C for 1 h twice. After heat-shock treatment, *Drosophila* were kept at 25 °C. These flies were dissected and observed at 10 days after clone induction.

## Genotyping unmarked fly lines

Non-lethal PCR genotyping was performed using adult wings, following previously described methods [94,95]. In summary, a pair of adult wings from individual *Drosophila* was collected and placed in PCR tubes. Next, 20 μL of PCR mix containing Platinum SuperFi DNA Polymerase (Invitrogen; Catalog #12351010) and amplification primers was added. The PCR products were subsequently analyzed using electrophoresis on a 1% agarose gel containing ethidium bromide.

## Paraquat, bleomycin, and tunicamycin treatment

Adult female *Drosophila* were transferred from medium to empty bottles for 2 h. The filter paper was cut into 3.5 × 6.0 cm pieces and treated with 5% (wt/vol) sucrose with 10 mM PQ, 50 μg/mL tunicamycin (TM), or 25 μg/mL BLM. Then, the moist papers were added to empty bottles for 24 h, and *Drosophila* were transferred into a new standard medium without PQ or TM. Identical *Drosophila* fed 5% sucrose were used as controls. PQ-, BLM-, or TM-induced damage experiments were performed in a temperature-controlled incubator at 25 °C, unless otherwise specified.

For RNAi or overexpression experiments with PQ, BLM, or TM treatment, the GAL4^ts lines and UAS lines were maintained at 18 °C to repress the Gal4 system. One day after eclosion, the adults were shifted to 29 °C to activate the Gal4 system, which induces RNAi or overexpression. Enclosed flies were incubated at 29 °C for 5 days. Then, *Drosophila* were starved in empty vials for 2 h and treated in 5% (wt/vol) sucrose with 10 mM PQ, 25 μg/mL BLM, or 50 μg/mL TM for 24 h, followed by recovery for 24 h. The controls were treated in the same way in parallel. The process of PQ, BLM, or TM treatment was performed at 29 °C.

## Immunostaining and microscopy analyses of *Drosophila* tissues

Guts were dissected in phosphate-buffered saline (PBS) and immersed in PBS with 4% EM-paraformaldehyde (PBS formula: 100 mM glutamic acid, 25 mM KCl, 20 mM $MgSO_4$, 4.5 mM $Na_2HPO_4$, 1 mM $MgCl_2$, pH 7.4) at room temperature (25 °C) for 45 min. Guts were washed in PBST (PBS, 0.1% Triton X-100) three times, 15 min each, followed by incubation with primary and second antibodies in PBST after soaking in BSA (PBS, 0.5% BSA, 0.1% Triton X-100) for 30 min. The primary antibodies are shown in Table 2.

Confocal images were acquired by a Leica TCS-SP8 confocal microscope. Images for each set of experiments were acquired as confocal stacks under the same settings, including 1,024 × 1,024 resolution and bi-direction scanning. The

**Table 2. List of antibodies.**

| Antibodies | Source | Use and Dilution |
|---|---|---|
| Rabbit anti-WRN | Abcam Cat# ab200 | IF 1:800 |
| Mouse anti-WRN(F-33) | Santa Cruz Cat# sc-101110 | WB 1:400 |
| Chicken polyclonal anti-GFP | Abcam Cat# ab13970 | IF 1:1000 |
| Rat monoclonal anti-mCherry | Invitrogen Cat# M11217 | IF 1:500 |
| Chicken anti-β-Galactosidase | Abcam Cat# ab9361 | IF 1:1000 |
| Rabbit anti-HA (C29F4) | Cell Signaling Technology Cat# 3724 | IF 1:1000 |
| Mouse anti-Armadillo | DSHB Cat# N2 7A1 | IF: 1:50 |
| Mouse anti-Delta | DSHB Cat# C594.9B | IF 1:100 |
| Mouse anti-Prospero | DSHB Cat# MR1A | IF 1:200 |
| Rabbit anti-phosphoHistone H3 (Ser10) | Millipore Cat# 06-570 | IF 1:1000 |
| Rabbit anti-p-eIf2α | Cell Signaling Technology Cat# 3597 | IF 1:200; WB 1:1000 |
| Rabbit anti-γ-H2AvD | Rockland Cat# 600-401-914 | IF 1:1000 |
| Mouse anti-β-tubulin | RayBiotech Cat# RM2003 | WB 1:1000 |
| Mouse anti-GAPDH | Beyotime Biotechnology Cat# AF0006 | WB 1:500 |
| Rabbit anti-pJNK (pTPpY) | Promega Cat# V7931 | IF 1:1000 |

gain value up reached 800 and offset was arranged from −0.1% to −0.2% in each image. To illuminate the Z-axis detail and optimize clarity, images were acquired as confocal z-stacks with numbers usually ranging from 5 to 10 and finally projected to one image. Adobe Photoshop CS5 and Adobe Illustrator 2021 were used to assemble the images. The midgut cells were counted using a Leica DM6-B microscope.

## Immunofluorescence for human colonic tissue analyses

Gut sections were incubated in xylene twice (15 min each) and dehydrated in 100% ethanol twice (5 min each). They were then dehydrated in 85% and 75% gradient ethanol (5 min each). After cleaning in distilled water, the sections were immersed in EDTA antigen retrieval buffer (pH 8.0) and maintained at a sub-boiling temperature for 8 min, left standing for 8 min, and maintained at a sub-boiling temperature for an additional 8 min. Sections were kept at room temperature to cool down and washed with PBS (pH 7.4) three times in a shaker (5 min each). After washing in flowing distilled water, sample sections were blocked in 0.5% BSA blocking solution for 30 min. Samples were then incubated with primary buffer at 4 °C overnight.

After washing with PBS three times (5 min each) in the shaker, the sections were incubated with the secondary buffer at 4 °C overnight, followed by incubation with DAPI solution at room temperature for 10 min in the dark.

After staining with DAPI, sections were mounted in mounting medium. Fluorescence images were obtained using the Leica TCS-SP8 confocal microscope.

## Gut lysate and western blotting analyses

Human colon tissues or isolated mouse crypts were mixed in RIPA Lysis Buffer (P0013B; Beyotime Biotechnology, Haimen, China) with appropriate protease inhibitors and then transferred to liquid nitrogen. The tissues were placed on ice 30 min after using a grinding rod for homogenization. The lysate supernatant was collected and total proteins were quantified using the PierceRapid Gold BCA Protein Assay Kit (A53226; Thermo). Each sample was supplemented with 5× loading buffer and boiled for 12 min. The protein gels were transferred to polyvinylidene difluoride (PVDF) membranes and blocked with 5% TBST with defatted milk for 1 h. After blocking, samples were incubated in the primary antibody overnight at 4 °C and washed with TBST three times at 25 °C (10 min each). PVDF membranes were incubated with the horseradish peroxidase-labeled secondary antibody in the dark and shaking about 2 h at room temperature.

Antibodies for western blotting are shown in Table 2. The secondary antibodies were horseradish peroxidase-conjugated Goat anti-mouse (Beyotime Biotechnology, #A0216, 1:1,000) and Goat anti-rabbit (Thermo, #A0208, 1:1,000).

## RT-qPCR

One hundred midguts were dissected from female flies per group in the experiment. Flies were dissected in 4 °C diethylpyrocarbonate (DEPC)-treated water-PBS, incubated in 1 mg/mL Elastase (Sigma, Cat. No. E0258) combined DEPC-PBS at room temperature, and mixed gently five times every 15 min. After dissociation, the samples were centrifuged at 400$g$ for 20 min at 4 °C, re-suspended in 4 °C DEPC-treated water-PBS, filtered with 70 mm filters (Biologix, Sao Paolo, Brazil). As the meanwhile, whole midguts from female flies were used to achieve total RNA using the TRIzol reagent (Thermo Fisher Scientific, Waltham, MA, USA) following the user guide. The PrimeScript RT Reagent Kit (TaKaRa, Kusatsu, Japan) was used to synthesize cDNA. RNA was reverse-transcribed by oligo dT. First-strand cDNA was diluted with sterile water 50 times and used for real-time PCR using the Quant-Studio 5 System (Thermo Fisher Scientific) and SYBRGreen (Genestar, Irvine, CA, USA). The reference standard group was Rp49, and expression levels were counted by the $2^{-\Delta\Delta CT}$ method. The expression level of the standard sample was normalized to 1.0. All primer sequences for qPCR are available upon reasonable request. Primer sequences used for qPCR are available upon request.

## RNA-Seq and data analysis

The *Drosophila* crosses were cultured on normal food at 25°C. *w*$^{1118}$ *Drosophila* were used as the control for *WRNexo-null mutant Drosophila* (*WRN*$^{10}$/*WRN*$^{12}$). *WRNexo*-null and wild-type flies were maintained at room temperature for homeostasis or injury assays. About 50 female flies for each biological replicate were dissected in PBS on ice. Midguts were frozen on drikold and isothiocyanate-alcoholphenyl-chloroform was used to collect total RNA after dissection. Then, total RNA was sent to Berry Genomics Corporation (Beijing, China) for sequencing on the NovaSeq 6000 platform (Illumina, San Diego, CA, USA). Quality control was performed using FastQC (version 0.11.8).

The raw RNA-seq data with read lengths of 150 bp were aligned to the *Drosophila* reference genome (Ensembl BDGP6 release-89). The aligned reads (sam files) were transferred to bam files and sorted using SAMtools. Gene expression levels in different samples were determined using DESeq2 (version 1.22.2). Differential expression was verified if $p \leq 0.05$ following Benjamini–Hochberg correction.

## Fluorescence intensity analyses

Immunofluorescence images were analyzed by confocal microscopy. The fluorescence intensity statistics in the region of interest (ROI) were calculated using ImageJ. The detailed process for the quantitative analysis of fluorescence intensity was described previously [97]. Briefly, the steps were as follows.

Open image: File -> Open.

Split Channels: Image -> Color -> Split Channels. Select the channel for calculation and strike out redundant channels.

Scale Setting: Analyze -> Set scale -> Click to remove scale. Set the scale in pixels.

Measurements Setting: Analyze -> Set Measurements. Choose the boxes "Area," "Integrated Density," and "limit to threshold." Click the "OK" box.

Choose the ROI or cell by different kinds of drawing/selection tools and select another smaller region around the ROI or cell as background.

Choose different types of ROI, cells, and background regions using "ROI Manager."

Select "Measure" and calculate the integrated density.

Integrated density = Integrated density of ROI or cell − Integrated density of background region/Area of background region × Area of ROI or cell.

## Gray value analysis of western blotting bands

A gray value analysis of bands obtained by western blotting was performed using ImageJ. The specific methods were as follows.

File -> Open. Choose the selected image.

Edit -> Invert.

Analyze -> Set Scale -> Choose the image scale to remove. Set the scale in pixel.

Analyze -> Set Measurements. Choose the bottom of "Area" -> "Mean Gray Value," limited to threshold, and choose the bottom of "OK."

Pick up the band region and choose a smaller area as the background.

Choose the ROI Manager to select the band and background regions.

Click the "Measure" box to compute the average gray value.

(Band gray value = (Mean gray value of band region × Area of band region) − (Background Mean gray value × Area of background region)).

## Lifespan assays

For survival tests under normal conditions, 100 female flies in three groups (2 days old) with the same phenotype ($w^{1118}$, $WRNexo^{10/15}$, and $WRNexo^{10/12}$) were collected and fed in a temperature-controlled incubator at 25 °C. For life span assays involving WRNexo overexpression, 50 female flies per group (2 days old, initially reared at 18 °C) were transferred to a temperature-controlled incubator at 29 °C to activate the UAS-Gal4 system. For life span assays involving WRNexo knockdown, 100 female flies per group (30 days old, initially maintained at 18 °C) were transferred to 29 °C. In all assays, the flies were monitored daily for survival, with the living flies transferred to fresh food tubes every 2 days.

## Cell fractionation and mass spectrometry

The midguts of *Drosophila* were rinsed and collected using chilled PBS. Subsequently, the nuclear and cytoplasmic components were isolated utilizing a Nuclear-Cytoplasmic Protein Extraction Kit (Beyotime, Shanghai, China, Catalog #P0028), following the protocols suggested by the manufacturer.

Proteins were separated by SDS-PAGE, and the gel bands were excised from the gel. For in-gel tryptic digestion, gel pieces were destained in 50 mM $NH_4HCO_3$ in 50% acetonitrile (v/v) until clear. Gel pieces were dehydrated with 100 µl of 100% acetonitrile for 5 min, the liquid removed, and the gel pieces rehydrated in 10 mM dithiothreitol and incubated at 56 °C for 60 min. Gel pieces were again dehydrated in 100% acetonitrile, liquid was removed and gel pieces were rehydrated with 55 mM iodoacetamide. Samples were incubated at room temperature, in the dark for 45 min. Gel pieces were washed with 50 mM $NH_4HCO_3$ and dehydrated with 100% acetonitrile. Gel pieces were rehydrated with 10 ng/µl trypsin resuspended in 50 mM $NH_4HCO_3$ on ice for 1 h. Excess liquid was removed and gel pieces were digested with trypsin at 37 °C overnight. Peptides were extracted with 50% acetonitrile/5% formic acid, followed by 100% acetonitrile. Peptides were dried to completion and resuspended in 2% acetonitrile/0.1% formic acid.

Peptides were dissolved in 0.1% FA (formic acid, Sigma), 2% CAN (Acetonitrile, Thermo), directly loaded onto a reversed-phase analytical column (75 umi.d.× 150 mm, packed with Acclaim PepMap RSLC C18, 2 um, 100Å, nanoViper). The gradient was comprised of an increase from 5% to 50% solvent B (0.1% FA, 80% ACN) over 40 min, and climbing to 90% in 5 min, then holding at 90% for the 5 min. The flow rate was 300 nl/min. The MS analysis was performed on QExactiveTM Mass Spectrometer (Thermo). The peptides were subjected to NSI source followed by tandem mass spectrometry (MS/MS) in Q ExactiveTM Mass Spectrometer coupled online to the HPLC. Intact peptides were detected in the Orbitrap at a resolution of 70,000 and peptides were selected for MS/MS using NCE setting as 27. The ion fragments were detected in the Orbitrap at a resolution of 17, 500. A data-dependent procedure that alternated between one MS

scan followed by 20 MS/MS scans was applied for the top 20 precursor ions above a threshold ion count of 1E4 in the MS survey scan with 30.0 s dynamic exclusion. The electrospray voltage applied was 2.0 kV. Automatic gain control was used to prevent overfilling of the ion trap and 1E5 ions were accumulated for the generation of MS/MS spectra. For MS scans, the $m/z$ scan range was 350–1,800 $m/z$. The fixed first mass was set as 100 $m/z$.

Protein identification was performed with MASCOT software by searching Uniprot Drosophila melanogaster. Searching Parameters are as follows:

Fixed modifications: Carbamidomethyl (C)

Variable modifications: Oxidation (M)

Enzyme: Trypsin

Maximum Missed Cleavages: 2

Peptide Mass Tolerance: 10 ppm

Fragment Mass Tolerance: 0.6 Da

Mass values: Monoisotopic

Significance threshold: 0.05

## Co-immunoprecipitation assay

For the Co-IP assay, pCDNA3.1-Flag or pCDNA3.1-Flag-Hsc70-3 and pCDNA3.1-HA-WRNexo were co-transfected into HEK293T cells seeded in a 10 cm cell culture dish. After 48 h of transfection, transfected cells were harvested, lysed using lysis buffer following the anti-FLAG M2 gel protocol (Cat# A2220; Millipore, Billerica, MA, USA). After the supernatant was collected, protein concentrations were determined using BCA kit. For IP, anti-FLAG M2 gel was added to the supernatant and rotated head-over-tail at 4 °C for 2 h. The beads were washed for 10 min (six times) with lysis buffer. Immunoprecipitated proteins and input samples were analyzed by western blotting after elution with 3× FLAG peptide.

## Pull-down assay

The transgenic plasmid of pet21a- Drosophila-WRNexo (aa127-aa357) and pGEX-6P-1-Drosophila-Hsc70-3 were constructed in-house. In brief, cDNA of WRNexo and Hsc70-3 were cloned into the pet21a and pGEX-6P-1 using the pEASY-Uni Seamless Cloning and Assembly Kit (CU101-02; TransGen Biotech, Beijing, China). The cDNA primers were as follows:

PET21a-Drosophila-WRNexo-F: ATGGGTCGCGGATCCGAATTCggtgctataaagtatt

PET21a-Drosophila-WRNexo-R: GTGGTGGTGGTGGTGCTCGAGgagaggaaatttact

GST-Drosophila-Hsc70-3/Bip-F: TTCCAGGGGCCCCTGGGATCCatgaagttatgcatatta

GST-Drosophila-Hsc70-3/Bip-R: CTCGAGTCGACCCGGGAATTCcagctcgtccttgagatc

His-tagged WRNexo and GST-tagged Hsc70-3/Bip were expressed in BL21 (DE3) bacterias and purified by using Ni-NTA agarose (Qiagen) or Glutathione agarose (Sigma), respectively. GST pull-down experiment was performed in the buffer containing 20 mM of Tris-HCI (pH 8.0), 90 mM of NaCl, 0.5 mM of EDTA, 1 mM of DTT, 10% glycerol, and 0.1% Tween. GST-tagged beads were washed for three times. (Washing buffer: 20 mM of Tris-HCI (pH 8.0), 120 mM of NaCl, 0.1% Tween.) Equal amounts of GST or GST-tagged Hsc70-3/Bip proteins were incubated with GST-tagged beads for 1 h at 4 °C. Then, his-tagged WRNexo (three times of GST-tagged proteins) were incubated with the element above for 1 h at 4 °C.

## ROS measurement with DHE

The ROS content was analyzed as described previously [34]. *Drosophila* midguts were dissected in Schneider's medium (HyClone) and transferred to 30 µM DHE (Invitrogen, Waltham, MA, USA) in a dark box for 5 min of incubation. Then, guts were washed by Schneider's medium three times and used to make slides to capture images as soon as possible. The confocal microscopy sections (568 nm excitation, 550–610 nm detection, fixed laser power, gain and offset settings) were used to obtain images. Signal intensities for the intestinal epithelium were measured using ImageJ, and ISCs were identified by *esg*-GFP expression.

## γ-Irradiation treatment

*Drosophila* were maintained at 25 °C and 60% humidity with a normal dark/light cycle. Adult flies (7 days) were collected in small collection discs, each containing 20 adult flies. As previous reports described [75], adult flies were irradiated in the γ-irradiation machine with a dose rate of 2.55 Gy/min for 5 Gy dose. After the irradiated flies were incubated for 1 h at 25 °C, the midguts were dissected for immunostaining.

## Statistical analyses

GraphPad Prism v7.0 was used for statistical analyses after verifying the normality and equivalence of variances. Data are presented as the means ± standard deviation from at least three independent biological replicates, unless otherwise specified. Statistical significance was determined using two-tailed Student's *t*-tests, unless otherwise specified. Significance is stated in the text or figure legends. For all tests, $p < 0.05$ was considered statistically significant.

## Software

R (version 3.5.3; downloaded from https://www.r-project.org/) was used for downstream analyses of RNA-seq data. Custom ImageJ was used for immunofluorescent staining and western blotting quantification (downloaded from https://imagej.nih.gov/ij/). Prism 7.0 (GraphPad) was also used (downloaded from https://www.graphpad.com/).

## Supporting information

**S1 Fig. WRNexo is expressed in *Drosophila* midguts and the expression increases with aging, related to** Fig 1**.** Underlying data and statistical analysis in S1 Data.
(DOCX)

**S2 Fig. WRNexo is essential for gut homeostasis, related to** Fig 2**.** Underlying data and statistical analysis in S2 Data.
(DOCX)

**S3 Fig. WRNexo functions cell-autonomously in ISCs to promote ISC proliferation, related to** Fig 3**.**
(DOCX)

**S4 Fig. WRNexo protein directly interacts with protein Hsc70-3/Bip, related to** Figs 4 **and** 5**.** Underlying data and statistical analysis in S4 Data.
(DOCX)

**S5 Fig. WRNexo regulates the cellular redox state and modulates ISC proliferation by UPR<sup>ER</sup> pathway, related to** Fig 6**.** Underlying data and statistical analysis in S6 Data.
(DOCX)

**S6 Fig.  WRNexo regulates ISC proliferation by Hsc70−3 and JNK signaling, related to** Fig 6. Underlying data and statistical analysis in S6 Data.
(DOCX)

**S7 Fig.  DNA double strands break induces WRNexo expression, related to** Fig 7. Underlying data and statistical analysis in S7 Data.
(DOCX)

**S1 Table.  List of gene counts from RNA-seq without BLM injury (WT and WRNexo-null), normalized by the DESeq2 R package, related to** Fig 4. Gene symbols are indicated.
(XLSX)

**S2 Table.  List of gene counts from RNA-seq with BLM injury (WT and WRNexo-null), normalized by the DESeq2 R package, related to** Fig 4. Gene symbols are indicated.
(XLSX)

**S3 Table.  Full Drosophila genotypes as they appear in each figure panel, related to** Figs 1–7 and S1–S7.
(XLSX)

**S4 Table.  Initial mass spectrometry (MASS) analysis was performed on nuclear proteins extracted from the midguts of flies carrying an esgts-GAL4-driven UAS-WRNexo-3xHA.**
(XLSX)

**S5 Table.  The second mass spectrometry (MASS) analysis of nuclear proteins extracted from the midguts of flies with esgts-GAL4-driven UAS-WRNexo-3xHA was performed.**
(XLSX)

**S6 Table.  The third mass spectrometry (MASS) analysis was performed on nuclear proteins extracted from the midguts of flies carrying an esgts-GAL4-driven UAS-WRNexo-3xHA.**
(XLSX)

**S1 Data.  Underlying data for** Figs 1 and S1.
(XLSX)

**S2 Data.  Underlying data for** Figs 2 and S2.
(XLSX)

**S3 Data.  Underlying data for** Fig 3.
(XLSX)

**S4 Data.  Underlying data for** Figs 4 and S4.
(XLSX)

**S5 Data.  Underlying data for** Fig 5.
(XLSX)

**S6 Data.  Underlying data for** Figs 6, S5, and S6.
(XLSX)

**S7 Data.  Underlying data for** Figs 7 and S7.
(XLSX)

**S1 Raw Images.**  **Raw Images for** Figs 1D, 1K, 5D, 5E, **and** 6J.
(PDF)

## Acknowledgments

We thank BDSC, VDRC, and TsingHua Fly Center for fly strains, and DSHB for antibodies.

## Author contributions

**Conceptualization:** Kun Wu, Juanyu Zhou, Haiyang Chen.

**Data curation:** Kun Wu.

**Investigation:** Kun Wu, Juanyu Zhou, Yiming Tang, Qiaoqiao Zhang, Xiaorong Li, Mei Luo, Yu Yuan, Xingzhu Liu, Zhendong Zhong.

**Methodology:** Kun Wu, Juanyu Zhou, Yiming Tang, Qiaoqiao Zhang, Lishou Xiong, Xiaorong Li, Zhangpeng Zhuo, Mei Luo, Yu Yuan, Xingzhu Liu, Zhendong Zhong.

**Resources:** Haiyang Chen.

**Software:** Zhangpeng Zhuo, Guanzheng Luo.

**Supervision:** Haiyang Chen.

**Validation:** XiaoXin Guo, Zihua Yu, Xiao Sheng, Guanzheng Luo.

**Visualization:** Qiaoqiao Zhang.

**Writing – original draft:** Kun Wu, Haiyang Chen.

**Writing – review & editing:** Kun Wu, Juanyu Zhou, Haiyang Chen.

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
