## [Editor Report · Decision Letter 0]

25 Mar 2024

Dear Dr Chen,

Thank you for submitting your manuscript entitled "Werner syndrome exonuclease promotes gut regeneration but causes ageassociated gut hyperplasia" for consideration as a Research Article by PLOS Biology.

Your manuscript has now been evaluated by the PLOS Biology editorial staff as well as by an academic editor with relevant expertise and I am writing to let you know that we would like to send your submission out for external peer review.

Once your full submission is complete, your paper will undergo a series of checks in preparation for peer review. After your manuscript has passed the checks it will be sent out for review. To provide the metadata for your submission, please Login to Editorial Manager (https://www.editorialmanager.com/pbiology) within two working days, i.e. by Mar 27 2024 11:59PM.

Kind regards,

Ines

--

Ines Alvarez-Garcia, PhD

Senior Editor

PLOS Biology

---

## [Decision Letter · Decision Letter 1]

16 Jun 2024

Dear Dr Chen,

Thank you for your patience while your manuscript entitled "Werner syndrome exonuclease promotes gut regeneration but causes ageassociated gut hyperplasia" was peer-reviewed at PLOS Biology. Please also accept my apologies for the delay in providing you with our decision. The manuscript has now been evaluated by the PLOS Biology editors, an Academic Editor with relevant expertise, and by three independent reviewers.

The reviews are attached below. As you will see, the reviewers find the conclusions interesting and novel, however they also raise several concerns that would need to be addressed. Reviewer 1 thinks that the functional interaction between Hsc70 and WRN exo should be follow up to strengthen the paper, and also points out that the DNA damage experiments are too superficial and don’t explain causal association, and that the use of colon samples is not relevant to the paper. In addition, this reviewer finds some of the conclusions overstated, and thinks they should be toned down. Reviewer 2 notes that the main experiments are performed with fly lines that are not backcrossed into a common background, and that such backcrossing is essential when looking at aging, as the background could induce false positive. The Academic Editor agrees, thus we would like you to repeat these experiments with backcrossed lines, as suggested by this reviewer. Reviewer 2 also asks for several clarifications regarding the role of Xbp1 activation and ageing, and, along with Reviewer 3, makes several suggestions to improve the experiments, including additional controls and clarifications.

In light of the reviews, we would like to invite you to revise the work to thoroughly address the reviewers' reports. Given the extent of revision needed, we cannot make a decision about publication until we have seen the revised manuscript and your response to the reviewers' comments. Your revised manuscript is likely to be sent for further evaluation by all or a subset of the reviewers.

We expect to receive your revised manuscript within 6 months. Please email us (plosbiology@plos.org) if you have any questions or concerns.

**IMPORTANT - SUBMITTING YOUR REVISION**

3. Resubmission Checklist

a) *PLOS Data Policy*

b) *Published Peer Review*

d) *Blurb*

Please also provide a blurb which (if accepted) will be included in our weekly and monthly Electronic Table of Contents, sent out to readers of PLOS Biology, and may be used to promote your article in social media. The blurb should be about 30-40 words long and is subject to editorial changes. It should, without exaggeration, entice people to read your manuscript. It should not be redundant with the title and should not contain acronyms or abbreviations. For examples, view our author guidelines: https://journals.plos.org/plosbiology/s/revising-your-manuscript#loc-blurb

Sincerely,

Ines

--

Ines Alvarez-Garcia, PhD

Senior Editor

PLOS Biology

Reviewers' comments

Rev. 1:

It is a worthwhile study to investigate the role of WRN in the aging process in the gut, but this is not as novel as they claim and there has been many studies of WRN expression with aging in humans and model systems.

There are many semantic errors in the text and it has not been proofed for English language.

The work represents an overload of diverse data that represent different aspects and do not all fit together. The quality of the experiments are generally high but too many diverse things are being explored and often not sufficiently in depth.

The observations of an association between Hsc70 and WRN exo (Fig 5) are potentially interesting, and they need to follow that up to determine if there is a functional interaction between these two proteins. If Hsc70 fragments stimulate the exonuclease activity in vitro, it would strongly support their finding.

The DNA damage experiments are somewhat superficial and do not explain causal association. Also they, use UV and claim it induces DNA double strand breaks. That is not the case.

The use of human colon samples is not so relevant here. As they point out, the Drosophila WRN only has Exo activity and is therefore quite different from the human, which also has helicase activity and there are many studies with human WRN implicating it in replication, but mainly via helicase function.

Rev. 2:

In this manuscript, Wu and coworkers examine the role of Werner syndrome exonuclease in natural ageing. This is an interesting topic as the loss-of-function in this exonuclease results in progeria but the role of the protein in normal ageing has not been examined. The authors present data, obtained mostly in Drosophila, to support their model where WRNexo accumulates due to DNA damage in intestinal stem cells (ISCs), modulates the ER unfolded protein response and redox balance in the ISCs and drives intestinal hyperplasia during ageing. In a way, this contrasts to the progeria observed upon loss-of-function of WRN. Overall, the work is interesting and well executed but there are a few aspects of their model that do not appear to fit; I find they do not make sense, at least not to me, and I think they need to be addressed before I can recommend the manuscript for publication in PLoS Biology. These are predominantly to do with the analysis of Xbp1's role:

1. Regarding Xbp1 activation: This is usually assessed by looking at splicing of the 1ry transcript, but the authors look at Xbp1 nuclear localization and in the conditions where Xbp1-EGFP is overexpressed. What is this Xbp1-EGFP construct? Is it reporting splicing? In this case this should be made explicit and consistent throughout the text. Additionally, it would be important to look at the splicing of the native transcript in at least one of their experiments, to avoid any potential artefacts of Xbp1-EGFP overexpression.

2. Regarding the function of Xbp1: It is surprising and inconsistent with the other results presented that Xbp1 knockdown improves proliferation in cells with a loss-of-function of WRNexo. As WRNexo is required and sufficient to induce proliferation and it presumably activates Xbp1, the converse would be expected, that WRNexo over-expression requires Xbp1 to induce proliferation.

3. Regarding Xbp1 and aging: the authors argue for a pro-aging role of WRNexo but this is somewhat inconsistent with the induction of Xbp1 in ISCs being able to promote organismal longevity and reduce the hyperplasia (e.g. Wang et al PLoS Genetics 2015; Li et al iScience 2024).

4. Bip is not only transcriptionally induced by Xbp1, it also regulates Xbp1 splicing (e.g. Kimata et al JCB 2004). This should be made clear throughout the text.

Additionally, I would expect the authors to address they key aspects of the following concerns:

5. The authors present WRNexo as driving aging. They do show that it is required and sufficient for age-related hyperplasia, however, would it not be more appropriate to test if over-expression or knockdown of WRNexo in adult ISC is able to modulate lifespan in a way the authors would expect?

6. It does not appear that any of the fly lines used were backcrossed into a common background. Such backcrossing is essential when looking at aging or age-related phenotypes (e.g. lifespan, pH3+ accumulation in old flies etc.) in the fruit fly as differences in background could introduce false positives (see Partridge and Gems Nature 2007). The key experiments will have to be repeated with backcrossed lines.

7. In several figures, "control" is not clearly defined. It should be defined in the caption (driver alone, or a control RNAi, or simply wild-type flies?).

8. When analyzing data obtained by signal quantification from confocal images, I think it would be important to account for the fact that signal from different cells within one individual will be correlated by using e.g. mixed effects linear models.

Rev. 3:

The authors present a study investigating the function of WRN in regulating ISC proliferation in response to aging. The further findings have significant implications that WRN can bind to Hsc70-3/Bip and regulate UPR. However, addressing the major concerns will strengthen the manuscript. To see specific comments below.

Major concerns:

1. In Figure 2C, the flies used are stated to be 30-day-old, whereas in Figures 2A, 2D, and 2E, the flies used are mentioned to be 45-day-old. To ensure accurate comparisons, it is recommended to use the same age groups for all figures.

2. Figures 2L and 2M lack a control group for comparison.

3. For Fig. 3, Q to T. the expression of WRNexo cDNA by esg-Gal4 rescued the proliferation defect of ISCs in WRNexo-null midguts after injury, but there are no Figs to show proliferation defect in WRNexo-null midguts.

4. The authors suggested that WRNexo contributes to the response to ER stress, therefore it should show that the ER stress can be rescued in response to TM treatment when WRNexo is blocked.

5. The authors hypothesize that the overexpressed WRNexo interacts with Hsc70-3, inhibiting its function and subsequently inducing Xbp1 splicing. However, to further support this hypothesis, additional experiments should be conducted.

6. In the last part of the result, the expression of WRNexo can repress DNA damage, however, the induction of DNA damage corresponded to an upregulation of WRNexo. The authors should provide explanations for supporting the findings.

Minor concerns:

1. The conclusion that "these data indicate that the depletion of WRNexo led to the loss of gut homeostasis upon aging in Drosophila" should be more specific.

2. "Following PQ treatment, ROS levels in esg+ cells of flies with esgts-Gal4-driven WRNexo RNAi expression were significantly lower than in the controls. (Fig. 6, I to K)". The authors should do more work for the similar grammar problems.

---

## [Decision Letter · Decision Letter 2]

6 Feb 2025

Dear Dr Chen,

Thank you for your patience while we considered your revised manuscript entitled "Werner syndrome exonuclease promotes gut regeneration but causes age-associated gut hyperplasia" for publication as a Research Article at PLOS Biology. This revised version of your manuscript has been evaluated by the PLOS Biology editors, the Academic Editor and the three original reviewers.

Based on the reviews (attached below), we are likely to accept this manuscript for publication, provided you satisfactorily address the remaining points raised by Reviewer 2. Please also make sure to address the data and other policy-related requests stated below.

In addition, we would like you to consider a suggestion to improve the title:

"Werner syndrome exonuclease promotes gut regeneration and causes age-associated gut hyperplasia in Drosophila"

We expect to receive your revised manuscript within two weeks.

*Published Peer Review History*

*Press*

Sincerely,

Ines

--

Ines Alvarez-Garcia, PhD

Senior Editor

PLOS Biology

Fig. 1C, E, F, I, J, L, M, P, Q, Z, A’, B’; Fig. 2A, H, I, J, M, Q, R, T, X, A’; Fig. 3A-D, T; Fig. 4A, B, F, J, G; Fig. 5K, P, U; Fig. 6C-F, I, K, O, R, W, X; Fig. 7C, D, G, H, M, N; Fig. S1E, F, H, I, J; Fig. S2F, J, K, O; Fig. S4A, B; Fig. S5E, H; Fig. S6B, D, H, M, Q and Fig. S7D, H

CODE POLICY

We require the original, uncropped and minimally adjusted images supporting all blot and gel results reported in an article's figures or Supporting Information files. Please provide the raw gels shown in the following figures:

Fig. 1K; Fig. 5D, E and Fig. 6J

We will require these files before a manuscript can be accepted so please prepare and upload them now. Please carefully read our guidelines for how to prepare and upload this data: https://journals.plos.org/plosbiology/s/figures#loc-blot-and-gel-reporting-requirements

Reviewers' comments

Rev. 1: Vilhelm Bohr - note that his reviewer has signed his review.

I think they have adequately responded to the reviewers concerns and that the MS should be acceptable.

Rev. 2:

I thank the authors for taking my comments on board. I think most have been addressed sufficiently well but there are two where I would appreciate the authors make small additional changes in the text.

Regarding my original comment (1) about the Xbp1 GFP reporter, it is clear even from the diagram provided by the authors in the rebuttal that this construct is reporting the splicing of Xbp1 and not simply its nuclear localisation. It would be good if the authors would make this clear on page 13. The qPCR shown in the rebuttal should also be shown in the manuscript (e.g. in a supplementary figure).

Regarding my comment (6) about backcrossing: thank you for clarifying. However, I find that the information in the methods is still not sufficient. Please explicitly note which lines were backcrossed or write that all lines used in the manuscript were backcrossed, and provide information about how the unmarked mutants/constructs were backcrossed (I believe the Crispr mutants were not marked, apologies if I have misunderstood).

Additional minor suggested changes:

Page 2, end of the abstract: "elevated WNRexo-mediated mechanism" - the expression is a bit clumsy - please change for clarity.

Page 6, start of the results: "contrary to our prediction" - please explain the prediction and reasoning for clarity.

Page 11, section title: "via regulate" - by regulating?

Page 12: Hsp70Ba, Hsp70Bb, Hsp68, Hsp67Ba and Hsp67Bc are referred to as UPR-ER genes - please provide a reference for this. If they are not UPR-ER genes (as I belive they are not ER chaperones), it may be advisable to refer to them as "involved in proteostasis" or something similarly broad.

Page 14, top of the page: "Xbp1 encodes a … signalling," - this is somewhat redundant.

Rev. 3: Aiguo Tian - note that this reviewer has signed his review.

My concerns have been addressed.

---

## [Editor Report · Decision Letter 3]

18 Mar 2025

Dear Dr Chen,

Thank you for the submission of your revised Research Article entitled "Werner syndrome exonuclease promotes gut regeneration and causes age-associated gut hyperplasia in Drosophila" for publication in PLOS Biology. On behalf of my colleagues and the Academic Editor, Alex Gould, I am delighted to let you know that we can in principle accept your manuscript for publication, provided you address any remaining formatting and reporting issues. These will be detailed in an email you should receive within 2-3 business days from our colleagues in the journal operations team; no action is required from you until then. Please note that we will not be able to formally accept your manuscript and schedule it for publication until you have completed any requested changes.

PRESS

Sincerely, 

Ines

--

Ines Alvarez-Garcia, PhD

Senior Editor

PLOS Biology
